# GPT IS BECOMING A TURING MACHINE: HERE ARE SOME WAYS TO PROGRAM IT

## ABSTRACT

We demonstrate that, through appropriate prompting, GPT-3 can be triggered to perform iterative behaviours necessary to execute (rather than just write or recall) programs that involve loops, including several popular algorithms found in computer science curricula or software developer interviews. We trigger execution and description of **iterations** by **regimenting self-attention** (IRSA) in one (or a combination) of three ways: 1) Using strong repetitive structure in an example of an execution path of a target program for one particular input, 2) Prompting with fragments of execution paths, and 3) Explicitly forbidding (skipping) self-attention to parts of the generated text. On a dynamic program execution, IRSA leads to larger accuracy gains than replacing the model with the much more powerful GPT-4. IRSA has promising applications in education, as the prompts and responses resemble student assignments in data structures and algorithms classes. Our findings hold implications for evaluating LLMs, which typically target the in-context learning: We show that prompts that may not even cover one full task example can trigger algorithmic behaviour, allowing solving problems previously thought of as hard for LLMs, such as logical puzzles. Consequently, prompt design plays an even more critical role in LLM performance than previously recognized.

## 1 INTRODUCTION

Large language models (LLMs) (Brown et al., 2020; Rae et al., 2021; Chowdhery et al., 2022; OpenAI, 2023) are trained on large text datasets, which typically include descriptions of procedures and even computer programs (Chen et al., 2021). Their performance on complex reasoning tasks remains limited even with advanced prompting methods, e.g. Chain-of-Thought (CoT) (Shwartz et al., 2020; Zelikman et al., 2022; Nye et al., 2021; Wei et al., 2022; Wang et al., 2022b; Zhou et al., 2022; Creswell et al., 2022; Wang et al., 2022a; Liu et al., 2022; Kojima et al., 2022; Li et al., 2022a). This implies that despite their size, current LLMs are unlikely to execute algorithms or solve problems such as logical deduction and logical grid puzzles in BIG-bench Lite (Srivastava et al., 2022), that require many (or iterated) reasoning steps in a direct, savant-like manner.

LLMs generate tokens in order, each based on previous tokens in the sequence, whether these are part of the prompt or have just been generated by the LLM itself. Such self-attention could allow an LLM to use all previously generated tokens as the store of information needed for tracking reasoning steps, states, etc.[1]. Such use of generated tokens would resemble a classical Turing Machine with its memory tape Turing (1936). In principle, a non-trivial recurrent transformer model with infinite attention could be Turing-complete and capable of executing arbitrary routines, as long as the attention mechanism can be controlled stringently enough. But, even in relatively simple settings, LLMs appear to resist strict controls. Slight changes in prompts can yield dramatically different responses Liu et al. (2021); Malkin et al. (2022); Shi et al. (2023), because many recurrent patterns in the training data are encoded into a single model, and learned patterns overlap and vary in the context size. Thus it is easy to mislead with a prompt with accidental alphabetical or numerical ordering, or some undetectable semantic bias (Zhao et al., 2021; Lu et al., 2022; Min et al., 2022).

In Section 2, we introduce much stricter attention controls that instruct LLMs to unroll reasoning steps of a procedure with the initially undetermined length, and decide when the solution is found:

---

[1]This is likely contributing to the success of CoT prompting, in addition to such prompts' explanatory value.

**Iteration by Regimenting Self-Attention (IRSA).** The basic way to achieve such deliberate self-attention control is through highly structured prompting with an example of execution path for one example, as illustrated for Bubble Sort algorithm in Prompt 1, which encourages an LLM to output not just the sorted sequence but also the swap count (response in Prompt A.1 in Appendix), which is a challenging task to solve in a savant manner. We further explore **fragmented prompting** which combines multiple fragments of execution paths, as well as the strategy of skipping parts of generated text when performing self-attention. We also discuss interpreter/compiler prompts that can translate an algorithm in a high-level programming language into an IRSA prompt that GPT-3 can execute.

We present results on a wide range of algorithms taught in computer science curricula and used to test software engineers in coding interviews, including string manipulations, dynamic programming, and stack operations in Section 3. Our findings point to broader applications for LLMs in software engineering and education Gao et al. (2022); Parisi et al. (2022); Schick et al. (2023); Mialon et al. (2023). More pressingly, they point out a critical issue in evaluating in-context learning of LLMs, suggesting that current evaluations may underestimate LLMs' abilities if prompts can combine natural language instructions with algorithmic iterative reasoning. The sensitivity of the performance to prompt design may be amplified by the iterative reasoning triggered by the prompt, which will then beg the question: If one LLM beats another on a task, is it simply because we have not found the right prompt for the second model? E.g., IRSA increases the performance of GPT-3 family on logical deduction puzzles from 32% to 76%. The discussion in the Appendix also includes an experiment with GPT-4 OpenAI (2023) on a well-known dynamic programming task showing that even the latest member in the family cannot consistently execute code without prompting in IRSA style.

## 2 Iteration by Regimenting Self Attention (IRSA): Explain like I'm ~~five~~ autoregressive

Prompt 1, triggering an execution of the Bubble Sort algorithm on an arbitrary input sequence, illustrates the basics of IRSA. For one input sequence, the prompt shows all state changes and *explains each change before it occurs*. The explanation is colloquial, but the structure of it is both rigid and repetitive, strictly regimenting the attention to the rules (corresponding to program instructions) and state changes. This strategy hardens the attention sufficiently to facilitate disciplined procedural reasoning, while leaving non-regimented content open to interpretation. (Sorting a sequence of 4 integers is demonstrated, but the same prompt can also be used to sort characters alphabetically or animals by size, and be applied to both shorter and longer input lists.)

IRSA could be thought of as an instance of Chain-of-Thought prompting. However, a significant distinction lies in the number of reasoning steps, which is limited and fixed in usual CoT applications, and the thorough annotation of steps in the order of reasoning, which is especially important in the treatment of conditionals: Instead of specifying the effect of a state change (swapping two elements), and then explaining why it was done (because the two were out of order), the 'why' is given first. While either order may be equally *explanatory* in prompt, the difference becomes evident in *generation*, when LLM attempts to follow the prompt's blueprint. If the explanation follows making a choice in the prompt, then the generation will follow the same pattern: make a cognitive leap to decide on a move, then rationalize that choice. In IRSA, instead, the reasoning comes first, and it is further segmented into substeps, so that new tokens inform the future choices as soon as possible: `Check if 2<3. Is it true?` triggers evaluation, and then generated next token `No` or `Yes` triggers copying the pattern from the prompt leading to swapping the elements (or not). Similarly, a new iteration is triggered by first recalling the value of the swap flag.

The structure of the prompt acknowledges the LLM's autoregressive nature, and does not require big reasoning leaps in generation. Instead the LLM is instructed to use the generated token stream as a memory tape that triggers the desired behaviour. Interestingly, as LLMs can make educated guesses on how to follow any recipe, one can instruct with various levels of detail. Here, the investigation of the swap flag happens after all pairs have been visited, as we expect that an LLM may infer how to do the same in generation. In contrast, in Prompt A.4 the state includes the iterator $i$, which is checked after each state transition to detect when the time for deciding on the next iteration has come.

Examples of basic IRSA for single loop programs can be seen in Prompts A.5 and A.6, and for double loop programs in Prompts 1, A.4, and 2. In each of these examples, a single prompt is provided for a task, which, when combined with a new instance of the task, trigger the execution of an iterative algorithm, with potentially an unknown number of iterations until the stopping condition is met.

**Prompt 1.** Bubble Sort: The prompt describes iterative state evolution, including counting swaps, and making the determination when to stop. Playground link (use with 0 temperature)

```
Problem: 2, 3, 1, 5
EXECUTION
    Prep
    Length of the list: 4
    Number of consecutive pairs: 3
    a=[2 3 1 5]
    set n_swaps=0
    EndPrep
    Iteration:
        set swap_flag=false. The state is:
        State: a=[2 3 1 5], n_swaps=0, swap_flag=false EndState
        Pair a[1,2] = [2 3] Check if 2<3. Is it true? Yes.
                            Because of that, we leave state as is
        State: a=[2 3 1 5], n_swaps=0, swap_flag=false
        Pair a[2,3]= [3  1] Check if 3<1.  Is it true? No.
                            Thus, we swap_flag=true, increase n_swaps by one,
                            and in the latest a=[2 3 1 5]
                            swap 3 and 1 to get into state:
        State: a=[2 1 3 5], n_swaps=1, swap_flag=true EndState
        Pair a[3,4]= [3 5]  Check if 3<5. Is it true? Yes.
                            Because of that, we leave state as is
        State: a=[2 1 3 5], n_swaps=1, swap_flag=true EndState
        swap_flag is true, so do another iteration
    Iteration:
        set swap_flag=false. The state is:
        State: a=[2 1 3 5], n_swaps=1, swap_flag=false EndState
        Pair a[1,2] = [2 1] Check if 2<1. Is it true? No.
                            Thus, we set swap_flag=true, increase n_swaps by one,
                            and in the latest a=[2, 1, 3, 5]
                            swap 2 and 1 to get into state:
        State: a=[1 2 3 5], n_swaps=2, swap_flag=true EndState
        Pair a[2,3] = [2 3] Check if 2<3. Is it true? Yes.
                            Because of that, we leave state as is
        State: a=[1 2 3 5], n_swaps=2, swap_flag=true EndState
        Pair a[3,4] = [3 5] Check if 3<5. Is it true? Yes.
                            Because of that, we leave state as is
        State: a=[1 2 3 5], n_swaps=2, swap_flag=true EndState
        swap_flag is true, so do another iteration
    Iteration:
        set swap_flag=false. The state is:
        State: a=[1 2 3 5], n_swaps=2, swap_flag=false EndState
        Pair a[1,2] = [1 2] Check if 1<2. Is it true? Yes.
                            Because of that, we leave state as is
        State: a=[1 2 3 5], n_swaps=2, swap_flag=false EndState
        Pair a[2,3] = [2 3] Check if 2<3. Is it true? Yes.
                            Because of that, we leave state as is
        State: a=[1 2 3 5], n_swaps=2, swap_flag=false EndState
        Pair a[3,4] = [3 5] Check if 3<5. Is it true? Yes.
                            Because of that, we leave state as is
        State: a=[1 2 3 5], n_swaps=2, swap_flag=false EndState
        swap_flag is false, so stop the iteration
Final List: 1, 2, 3, 5
Number of swaps: 2
END OF EXECUTION

Problem: 0, 3, 8, 5, 6
EXECUTION
```

**Prompt 2.** Logical deduction puzzle IRSA prompt. Playground link (use with 0 temperature)

```
PUZZLE: The following objects need to be ordered. obj1 is the biggest. obj2 is smaller than obj3.
obj1 is bigger than obj2. QUESTION: Which object is the biggest?
START
Parsing step:
    Items: obj1, obj2, obj3
    Numbe of items: 3
    Statement 1: obj1 is the biggest.
    Statement 2: obj2 is smaller than obj3.
    Statement 3: obj1 is bigger than obj2.
Scoring identification step:
     Scores will refer to size.
     Since we have 3 items, let's assume that the biggest gets a score  of 3 pounds
     and the smallest gets the score of 1 pound.
Translation step:
    Available variable names: x, y, z, a, b, c
    Map item scores of 'obj1', 'obj2', 'obj3' to variable names x, y, z
    obj1 score is x; obj2 score is y; obj3 is z;
    Statement 1: 'x' is the biggest.
    Statement 2: 'y' is smaller than 'z'.
    Statement 3:  'x' is bigger than 'y'.
Initialization step:
    Words used to qualify the realtionsips: smaller, bigger, biggest
    Orientation step:
        the biggest: refers to the score of 3
        smaller: refers to smaller score
        bigger:  refers to larger score
    Initialize so that all scores are  different numbers between 1 and 3
    Score_assignment_A:
    x=2, y=3, z=1
Iterative reasoning
Iteration 1:
    update_flag=false
    Statement 1:  'x' is the biggest, meaning: x should be 3
    In Score_assignment_A, x is 2
    x is not what it should be, so we need to make a change, so we set update_flag=true and we need to make a swap.
    In the statement there is only one variable and it is  x. We need  to find another. We want x to be 3,
    but we see that in Score_assignment_A that 3 is assigned to y, so we swap  values of x and y to make
    Score_assignment_B:
    x=3, y=2, z=1
    Statement 2: 'y' is smaller than 'z', meaning: y<z
    In Score_assignment_B, y is 2 and z is 1,  so y<z maps to 2<1
    2<1 is false, so we need to make a change, so we set update_flag=true and we  need  ot make a swap.
    In the statement there are two variables and those are y and z so we swap in Score_assignment_B to make
    Score_assignment_C:
    x=3, y=1, z=2
    Statement 3: ' x' is bigger than 'y', meaning x>y
    In Score_assignment_C, x is 3 and y is 1,  so x>y maps to 3>1
    3>1 is true, so we don't need to make a change.
End of iteration. Since update_flag is true, we need more iterations.
Iteration 2:
    update_flag=false
    Statement 1:  'x' is the biggest, meaning: x=3
    In Score_assignment_C, x is 3,  so x=3 maps to 3=3
    3=3  is true, so we don't need to make a change.
    Statement 2: 'y' is smaller than z, meaning: y<z
    In Score_assignment_C, y is 1 and z is 2, so y<z maps to 1<2
    1<2 is true, so we don't need to make a change.
    Statement 3: 'x' is bigger than y, meaning x>y
    In Score_assignment_C, x is 3 and y is 1,  so x>y maps to 3>1
    3>1 is true, so we don't need to make a change.
End of iteration. Since update_flag is false, we have finished all iterations and found the correct order.
The correct score assignment is the last (Score_assignment_C):
x=3, y=1, z=2
Reverse translation step:
Map items 'obj1', 'obj2', 'obj3' to variable names x, y, z
so we replace x by obj1, y by obj2, and z by obj3 to get size scores:
obj1 has the score 3; obj2 has the score 1; obj3 has the score 2

Question: Which object is the biggest?
Answer: obj1
Sorting all by score starting with obj1:
with score 3, obj1
with score 2, obj3
with score 1, obj2
END

PUZZLE: On a shelf, there are five books: a gray book, a red book, a purple book, a blue book, and a black book.
The red book is to the right of the gray book. The black book is to the left of the blue book.
The blue book is to the left of the gray book. The purple book is the second from the right.
QUESTION: Which is leftmost?
START
```

## 2.1 USING IRSA TO REASON OVER LOGICAL PUZZLES.

In addition to program execution, iterative reasoning is required in solving a number of NLP word prolems, (e.g., Srivastava et al. (2022)). The BIG-bench Logical Deduction task requires ordering several objects given their pairwise relationships in natural language (e.g., a robin is standing on a branch to the right of a raven, but a sparrow is the left-most). Even for a small number of objects, LLMs struggle to solve such puzzles in zero- or few-shot settings, much like how human solvers cannot just see the correct answer instantly without scratch paper. This task is not solved well by LLMs without external search/reasoning/inference algorithms, such as ThinkSum Ozturkler et al. (2023). However, a variant of BubbleSort algorithm adapted to this problem and shown in Prompt 2 can be used to solve 76% of these puzzles. The prompt first translates the problem into a canonical form, and then, in IRSA style, describes an iterative swapping procedure that rearranges the objects.

## 2.2 FRAGMENTED PROMPTING.

Another way to trigger iterative behaviour is through fragmented prompting, illustrated in Prompt 3), and which relies on **complete state specification** and **fragmentation.** Prompt 3 does not fully cover the entire execution path of any single example. Instead, it follows the first three state changes[2] for the sequence $2, 3, 1, 5$, and then stops in the middle of a sentence. Then it shows 6 additional fragments of execution paths for *different* problems.

Interestingly, this prompt triggers iterative behaviour, where the language model accurately executes the algorithm on a given input and outputs `END OF EXECUTION` when the termination condition is met. Viewing this prompt as an instance of in-context learning, it is challenging to classify it in usual terms. It goes beyond 0-shot learning as it contains explanations specific to the algorithmic sorting task. Yet, as opposed to what the few-shot CoT prompting might do, it does not work out any single example of array sorting. Instead, it provides fragments of patterns that can be stitched together to execute the algorithm (and GPT-3 CODE-DAVINCI-002 does execute it correctly for new inputs).

The potential advantage of such fragmented prompting is that the prompt can be shorter and include a greater variety of situations that may be encountered in new problems. A potential disadvantage is that the language model may get confused by the fragmentation and start hallucinating new independent fragments. In this case, we managed to avoid that by having the first fragment starting from the start of execution, going through several state transitions, and ending mid-sentence. Because of this, when a new problem is given, the language model starts running the execution path from the beginning, and later refers to various cases in the prompt for guidance on how to proceed.

## 2.3 SKIP ATTENTION.

Prompt 3 also illustrates the idea of attention skipping. Whether using a single-execution or a fragmented prompt, if the state in the `<state>*</state>` structure is complete, the attention mechanism can generate the next token without attending to all the generated text. **It is sufficient to attend to the prompt and the text generated after and including the last state.**

If the skipping is implemented on the server side, akin to stop word functionality, then skipping unnecessary attention saves computation: The state of the model at the end of the prompt is cached and used to continue processing from the latest generated `<state>` marker, ignoring the text generated in-between. Skip-to-state can also be implemented on the client side, iteratively updating the original prompt by concatenating the latest `<state>*</state>` structure to the original prompt and calling the generative model with `</state>` as a stop sequence (We did the latter in our experiments). In both cases, the skip-to-state strategy should increase the number of tokens that can be generated, as self-attention, which grows linearly with the generated text, is the primary cause for the token limitations. Skip-to-state strategy keeps the self-attention cost constant. As IRSA requires the unrolling of potentially long iterative algorithms, these savings are important. For example, running a dynamic program that keeps track of 2D matrices is only practical in this manner. (See also Schuurmans (2023) on an external memory approach to dealing with limited attention length. Here we deal with it by skipping parts of generated text, instead). Another advantage of skip-to-state attention is that by only attending to the necessary information, the generative model is less likely to

---

[2]The full execution path in this style is shown in Prompt A.4.

**Prompt 3. Fragments**: An incomplete path for the first few Buble Sort state transitions for one sequence is followed by state transitions involving *different* sequences at *different* execution points. Initial part of the response is marked green. **Skip attention**: The part of the response up to the last state is not needed to continue the generation. Only the prompt, the last `<state>*</state>`, and the text after it are necessary to generate the next token. Playground link (use with 0 temperature)

```
Problem: 2, 3, 1, 5
EXECUTION
    Length of the list: L=4
    Number of pairs: P=3
    a=[2 3 1 5]
    set n_swaps=0. set i=P=3. set swap_flag=true.
        <state> a=[2 3 1 5] i=3 P=3 n_swaps=0 swap_flag=true </state>
    Since i=3 and P=3, i and P are equal, so this iteration is done, but swap_flag is true,
    so we need another iteration
    Iteration:
        set swap_flag=false.  set i=0. The state is:
        <state> a=[2 3 1 5] i=0 P=3 n_swaps=0 swap_flag=false </state>
        Since i=0 and P=3, these two are different, so we continue
        a[i]=a[0]=2 a[i+1]=a[1]=3
        Because 2<3 is true we keep state as is and move on by increasing i
        <state> a=[2 3 1 5] i=1 P=3 n_swaps=0 swap_flag=false </state>
        Since i=1 and P=3, these two are different, so we continue
        a[i]=a[1]=3 a[i+1]=a[2]=1
        Because 3<1 is false we set swap_flag=true,increase n_swaps by one, and in a=[2 3 1 5]
        swap 3 and 1, and increase i, and keep P as is to get
        <state> a=[2 1 3 5] i=2 P=3 n_swaps=1 swap_flag=true </state>
        Since i=2 and

        <state> a=[6 5 8 9 1 2] i=2 P=5 n_swaps=5 swap_flag=false </state>
        Since i=2 and P=5 i and P are different, so we continue
        a[i]=a[2]=8 a[i+1]=a[3]=9
        Because 8<9 is true we we keep state as is and move on by increasing i
        <state> a=[6 5 8 9 1 2] i=3 P=5 n_swaps=5 swap_flag=false </state>

        <state> a=[9 1] i=0 P=1 n_swaps=2 swap_flag=true  </state>
        Since i=0 and P=1 i and P are different, so we continue
        a[i]=a[0]=9 a[i+1]=a[1]=1
        Because 9<1 is false we set swap_flag=true,increase n_swaps by one, and in a=[9 1] swap 9 and 1
        and increase i, and keep P as is to get
        <state> a=[1 9] i=1 P=1 n_swaps=3 swap_flag=true  </state>

        <state> a=[6 7 3 5] i=3 P=3 n_swaps=7 swap_flag=false </state>
        Since i=3 and P=3 i and P are equal, so this iteration is done, swap_flag is false, so stop
Final List: 6, 7, 3, 5
Number of swaps: 7
END OF EXECUTION
        <state> a=[3 5 6 8] i=3 P=3 n_swaps=1 swap_flag=true  </state>
        Since i=3 and P=3 i and P are equal, so this iteration is done, but swap_flag is true,
        so we need another iteration
    Iteration:
        sset swap_flag=false.  set i=0. The state is:
        <state> a=[3 5 6 8] i=0 P=3 n_swaps=1 swap_flag=false </state>

        <state> a=[2 8 1 3 5 7 4] i=1 P=6 n_swaps=5 swap_flag=false </state>
        Since i=1 and P=6 i and P are different, so we continue
        a[i]=a[1]=8 a[i+1]=a[2]=1
        Because 8<1 is false we set swap_flag=true,increase n_swaps by one, and in a=[2 8 1 3 5 7 4]
        swap 8 and 1 and increase i, and keep P as is to get
        <state> a=[2 1 8 3 5 7 4] i=2 P=6 n_swaps=6 swap_flag=true  </state>

        <state> a=[4 8] i=0 P=1 n_swaps=7 swap_flag=true  </state>
        Since i=0 and P=1 i and P are different, so we continue
        a[i]=a[0]=4 a[i+1]=a[1]=8
        Because 4<8 is true we we keep state as is and move on by increasing i
        <state> a=[4 8] i=1 P=1 n_swaps=7 swap_flag=true  </state>
Problem: 3, 1, 8, 9, 6
EXECUTION
    Length of the list: L=5
    Number of pairs: P=4
    a=[3 1 8 9 6]
    set n_swaps=0. set i=P=4. set swap_flag=true.
        <state> a=[3 1 8 9 6] i=4 P=4 n_swaps=0 swap_flag=true </state>
        Since i=4 and P=4 i and P are equal, so this iteration is done, but swap_flag is true,
        so we need another iteration
    Iteration:
        set swap_flag=false.  set i=0. The state is:

        <state> a=[3 1 8 9 6] i=0 P=4 n_swaps=0 swap_flag=false </state>
        Since i=
```

get confused by accidental patterns created in its own generated text. (See more in Section A.3 and Figure A.2.)

## 2.4 GPT AS A MACHINE LANGUAGE: PROMPTING TO INTERPRET/COMPILE A PROGRAM.

A general-purpose computer can execute algorithms that convert the text of a program into its machine code. Analogously, we designed IRSA prompts that turn code in some language into an execution path that can then be used in prompting (Section A.1). We used a "GPT compiler" for an invented programming language in Prompt A.2 to generate an IRSA-like execution path for the double-loop DP algorithm for the longest common subsequence problem, providing an LCS IRSA-prompt.

## 3 EXPERIMENTS

Our experiments include the following evaluations:

- **Basic IRSA**: Prompting with highly structured single execution path examples (Table 1). As opposed to CoT prompts providing multiple steps of reasoning shown for a few examples, IRSA prompts use single example designed to trigger iterative reasoning that is repeated until the stop condition is reached and the solution is found, and the execution path example for each task is deliberately chosen to be out-of-distribution (e.g., the Bubble Sort prompt features a worked-out example of sorting a four-number sequence in just three passes, while the dataset consists of five-number sequences requiring 2 to 5 iterations and up to 20 state transitions, with varying complexity across problems). Thus in terms of information they provide, these prompts can be seen as somewhere between single-shot and zero-shot prompts.
- **Skip-to-state IRSA**: Prompting as above, but with additional forced attention skipping. In this approach, the LLM is forced to attend only to the prompt and the last generated state as it iterates through the input to find the solution (illustrated at the end of Prompt 3). We also evaluate fragmented prompts (Table 2), where the prompt does not consist of a single complete execution path for an example, but instead shows several state-to-state transitions for different inputs.

- **Interpretation of new code.** As discussed in Sections 2.4, A.1, IRSA style prompting can take code in a high level language as the input and produce IRSA-like annotated execution paths, which will then also include the result of the execution in the end. We compare IRSA with the few-shot prompts in Nye et al. (2021) on interpreting and executing 100 synthetic Python programs (Table 3).

**Baselines.** To make fair comparisons and avoid unnecessary recomputation, we reused existing baselines from Srivastava et al. (2022) wherever possible, denoted by an asterisk (*): Logical deduction, Balanced parenthesis, and Longest common subsequences for long sequences. We created our own datasets and ran baselines for the following tasks: Bubble sort, Longest substring without repeating characters, and Longest common subsequence for short sequences. We include the best result from Srivastava et al. (2022) for the GPT family, as our experiments were mainly conducted using GPT-3. Our baselines included zero or few (5) shot prompting with or without relevant code added to the description of the task in the prompt (e.g. Prompt A.11). Few shot baselines were made with 5 different random choices of examples to be included in the prompt. The 'Guessing' strategy refers to picking the most frequently correct answer for a given task as a guess for each problem in the task, which is different from truly random guessing. Few-shot prompting could prime the answers to pick the most frequently seen answer, even when no understanding of the problem occurs, which makes our 'Guessing' strategy more reflective of the task difficulty.

**Models.** We have briefly experimented with different members of the GPT-3 family, but ran complete experiments with CODE-DAVINCI-002 for two reasons: TEXT-DAVINICI-002 and 003 often produced qualitatively similar results, and experimentation with the CODE-DAVINCI-002 was easier due to better combination of token quota and availability. Having been tuned on code, this model may have slight advantages over models tuned for more natural language tasks. Nevertheless, as we show in the experiments and discuss in Section A.3, without IRSA, CODE-DAVINCI-002 cannot solve the problems discussed here, even when it can generate the code that could. To induce iterative reasoning in LLMs, it appears that attention needs to be highly regimented through strong structure, and possibly additional attention control, such as the skip-to-state strategy we described in Section 2.3. This also applies to GPT-4 OpenAI (2023): In Section A.3.3 in Appendix, we show that prompting GPT-4 with straight-forward Prompts A.12, A.13, A.14 does not match the performance of IRSA in GPT-3.

**Datasets.** We test on a mix of reasoning tasks and challenging programming tasks included in computer science curricula and coding interviews for software engineers:

Table 1: IRSA compared with in-context learning baselines, and with the strategy of always guessing the most frequent answer. (*) denotes the best result for GPT-3 from the BIG-bench.

| Task | IRSA | Baseline | Guessing |
|---|---|---|---|
| Bubble sort | | | |
| - Prompt 1 | 0.74 | 0.27 | 0.23 |
| - Prompt A.4 | 1.00 | 0.27 | 0.23 |
| Longest substring | 1.00 | 0.60 | 0.59 |
| Logical deduction | 0.76 | 0.32* | 0.2 |
| Parentheses | 0.96 | 0.56* | 0.5 |

**Bubble sort.** We created a dataset of 100 random non-repeating digit sequences of length 5. The task is to predict the number of swaps needed to sort the sequence.

**Longest substring without repeating characters.** A classical coding interview question: Given a string of letters, find the length of the longest contiguous substring such that no letter appears more than once. We created a dataset of 100 random strings of length 7.

**Logical deduction Srivastava et al. (2022).** We include this task (Section 2.1) in experiments to emphasize the broad importance of triggering iteration in LLMs responses. Enabling LLMs to execute iterative algorithms through effective prompting could help solve numerous reasoning problems. In particualr, this task that involves solving a puzzle about an order of items/objects/persons, such as books on the shelf, birds on a branch, cars, golfers, etc., given several clues. We focus on a subtask involving 5 items, with varying sets of items and the types of ordering across the puzzles. While in-context learning with LLMs consistently solves less than 35% of puzzles, a recent combination of GPT-3 and probabilistic reasoning Ozturkler et al. (2023) was able to solve 77% of the puzzles. We reach a similar performance through IRSA, *without* an additional external reasoning mechanism.

**Valid parentheses Srivastava et al. (2022)** from the cs-algorithms challenge in BIG-bench. The goal is to evaluate LLMs ability to perform reasoning equivalent to the classical stack manipulations needed to check if a sequence of parentheses of different types is balanced. LLMs (including GPT) tend to do the same as chance (50%), except for PaLM with 3 shots, which gets around 75% accuracy.

**Longest common subsequence (long) Srivastava et al. (2022)** from the BIG-bench cs-algorithms challenge involves solving a classical dynamic programming problem. Defining a subsequence to be a sequence of symbols one could get by skipping arbitrary stretches in the original sequence, the task is to find the length of the longest subsequence common to two given sequences. LLMs do not do much better than chance on this task (∼10%).

**Longest common subsequence (short).** We created this dataset in the same manner as the above one, but limiting sequence lengths to be at most 6. This allows us to evaluate IRSA on more cases, ensuring it does not run out-of-memory (tokens) in generation .

**Synthetic Python programs.** We generated and evaluated 100 random programs involving arithmetic operations and (possibly nested) while and if statements as in Bieber et al. (2020); Nye et al. (2021).

**Basic IRSA results.**   Summary is provided in Table 1. In Bubble Sort evaluations we show the results using Prompt 1 (74%), and Prompt A.4 (100%). The latter tracks the full state including a loop iterator. Note that while the execution path for the prompt example $2, 3, 1, 5$ requires 3 iterations of the outer loop and 3 iterations in each inner loop, the dataset, with sequences of length 5, requires four iterations in the inner loop and a variable number of iterations of the outside loop – anywhere from 2 to 5 – and yet the model can execute the correct number of iterations based on the stoppage criterion. For the logical deduction puzzles, we used Prompt 2. even though the iterative reasoning logic there is faulty as it may enter an infinite loop. When that happens, the generation runs out of tokens and we simply used the answer after the 4th iteration in evaluation. Section A.3 suggests the potential for creating more effective prompts. Nevertheless, this prompt still leads to state-of-the-art results, comparable only with Ozturkler et al. (2023), which uses an external reasoning mechanism. The longest substring without repeating characters problems is solved with IRSA Prompt A.5 explained in Section A.2). To address the parentheses problem, we used Prompt A.6 in Section A.2.1.

Table 2: IRSA with skip-attention, Bubble Sort and Longest Common Subsequence problems. Fragmented prompting, Bubble Sort problems. (*) denotes the best GPT result in BIG-bench

| Baselines | Bubble Sort | LCS-S | LCS-L |
|---|---|---|---|
| 0-shot | 0.20 | 0.09 | $0.14^*$ |
| 0-shot + code | 0.20 | 0.11 | - |
| few shot | $0.25_{\pm 0.05}$ | $0.07_{\pm 0.01}$ | $0.16^*$ |
| few shot + code | $0.23_{\pm 0.03}$ | $0.06_{\pm 0.02}$ | - |
| Guessing | 0.23 | 0.44 | 0.10 |
| **IRSA skip-to-state** | | | |
| single path | 0.95 | 0.93 | 0.28 |
| 7 fragments | $0.99_{\pm 0.02}$ | - | - |
| 13 fragments | $0.97_{\pm 0.03}$ | - | - |
| 19 fragments | $0.99_{\pm 0.02}$ | - | - |
| 25 fragments | $0.97_{\pm 0.03}$ | - | - |

Table 3: Interpretation of 100 synthetic Python programs with arithmetics, *if* clauses and nested loops

| Interpreter Prompts | 1-shot | 2-shot | 3-shot |
|---|---|---|---|
| Execution trace in Nye et al. (2021) | 0.55 | 0.54 | 0.59 |
| IRSA | 0.85 | 0.86 | 0.91 |

**Skip-to-state attention results.** The longest common subsequence (LCS) problem requires a state including a $M \times N$ matrix with solutions for all prefixes of the two sequences of lengths $M$ and $N$. Without skip-to-state attention (Section 2.3), the API calls can run out of tokens. Using the approach described in Section 2.4, A.1, we compiled an execution path in Prompt A.3, and then used it to induce IRSA on LCS short (LCS-S) and LCS long (LCS-L) problems. Even with skip attention, the state was too large to fit the token limit for most of the problems in LCS-L from BIG-bench. Yet, IRSA with skip attention still beats the state-of-the-art significantly (Table 2). On shorter problems in LCS-S, where IRSA with skip-attention does not run out of tokens, the performance was a respectable 93%. Note that GPT-4, without IRSA, only has 69% accuracy on LCS-S (Section A.3.3).

We tested **fragmented prompting** of Bubble Sort execution (Table 2). For each selected number of fragments – 7, 13, 19, 25 – at least one of five randomly generated prompts achieved 100% accuracy. These prompts followed the format in Prompt 3, starting with the few state transitions from the beginning for the sequence $[2, 3, 1, 5]$ and then listing additional 6, 12, 18, or 24 fragments. Bubble Sort has 6 different transitions, and fully balanced instruction listing one, two, three, or four of each type, with a random sequence in the state, leads to a slightly better performance than having a completely randomly chosen execution path fragments. These six basic transitions, illustrated in Prompt 3, involve two ways of ending an iteration depending on the swap flag and four ways of changing the state: two possibilities for inequality being true or not, combined with two possible previous values of the swap flag. We found that the prompt sensitivity causes different prompts to fail for different test cases: Each of the fragmented prompt collections yields 100% as an ensemble.

**Interpretation of random programs.** Table 3 compares the scratchpad prompts in Nye et al. (2021) (Prompt A.8) – which show execution traces for three programs, but without the reasoning logic for state transitions and *if* and *while* triggered jumps – with the corresponding IRSA-style prompts (Prompt A.9) on interpretation of 100 Python programs. (Section A.1).

## 4 CONCLUSION

We demonstrated that GPT-3 can be triggered to execute iterative algorithms, including double loops, with variable termination conditions. This has consequences discussed in Appendix (Section A.3). For example, IRSA may find applications in sofware engineering and education. If LLMs are programmable (in addition to being natural language translators and analyzers), their evaluation probably needs to be rethought, esp. in cases where models are expected to make inferences for which we have algorithms, because in-context learning would cover prompts designed to execute them (Section A.3). Regimenting self-attention for a given task may require a level of effort (Section A.3.2, but even GPT-4 cannot execute programs consistently without IRSA (Section A.3.3).

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

## A APPENDIX

### A.1 GPT AS A MACHINE LANGUAGE: PROMPTING TO INTERPRET/COMPILE A PROGRAM.

A general-purpose computer can execute algorithms that convert the text of a program into its machine code. Analogously, we can design prompts with instructions on how to turn code in some language into execution paths that can then be used in prompting.

An example is shown in Prompt A.2 (Appendix), where several examples of hypothetical syntax for transforming states are given, including setting values of variables and matrices, printing them,

a single loop program execution, and the `detailed_max` function that breaks down steps and explains them. Then, the double loop dynamic programming algorithm for finding the longest common subsequence (LCS) is also presented in this new language. This prompt successfully triggers the correct execution of the algorithm, complete with detailed explanations and state transitions (green shaded in Prompt A.3). This can then be used as a prompt to execute the LCS algorithm on arbitrary inputs (Section 3). We should note that GPT-3 is still sensitive to small alterations in text, and Prompt A.2 does not always lead to good interpretations of the algorithm. The performance may depend on accidental deceptive patterns and inconsistencies in the prompt, as well as the input. Nevertheless, once the output has been verified as correct, the Prompt A.2 together with the response in Prompt A.3 became the prompt – IRSA 'machine code' for GPT — to execute (mostly correctly) the LCS algorithm for new inputs, as long as they are appended in the same format:

```
LCS:
Input: <seq1> <seq2> End of input
LCS Prep:
```

Most of the main paper is concerned with executing IRSA prompts, assuming that such prompts are written correctly for a given program to be executed on arbitrary inputs. I.e. IRSA is tested as *a way of programming* GPT, which may be involved in prompting techniques for reasoning problems, or an entry point to procedural instructions for those who are not trained as programmers (yet). The experiments are also meant to demonstrate that GPT is capable of disciplined execution of a given program.

If instead we want to turn a high-level programming language into IRSA for some input – which will then contain the output at the end, too – the testing wold need to involve interpretation of many programs given in that programming language, as attempted in Bieber et al. (2020); Nye et al. (2021). To demonstrate that IRSA is a good programming strategy to writing its own compilers, we chose the task studied in Nye et al. (2021), where random synthetic python programs and their execution traces were used to train and evaluate transformer models. As we do not tune models, but study the instruction strategies, we compare with the in-context learning task proposed in Nye et al. (2021). They used a 'scratchpad prompt' (their Appendix C) with three examples of Python code with its execution trace to trigger generation of similar traces for synthetic Python programs. We used the same three examples and modified the prompt with an IRSA-style explanation between each line and state pair in the trace so that it follows the IRSA recipe, but is still in the few-shot format, rather than in gradual and fragmented format of Prompt A.2.

As in Nye et al. (2021), we generated random Python programs following the recipe in Bieber et al. (2020) in their Supplemental material B, Figure 6, to include arithmetic manipulation of the input variable $v0$ as well as if statements involving it, and the (possibly nested) while loops with additional counter variables with randomly chosen names from $v1$-$v9$. We additionally limited the complexity of the dataset to include 0-2 control structures, although those structures could be consecutive or nested and they need not be the same. Here is a sample program:

```
def f(v0):
    v0 -= 0
    v0 *= 1
    v2 = 1
    while (v2 > 0):
        v2 -= 1
        v0 -= 2
        if (v0 % 10 < 3):
            v0 += 3
            v0 += 2
        v0 -= 3
    v0 += 3
    return v0

output = f(9)
```

We tested scratchpad (Prompt A.8) and its IRSA version (Prompt A.9) with the first, the first two, and all three examples described in Appendix C of Nye et al. (2021). As we see in Table 3, in terms of

generating the correct output for 100 random programs, the number of shots affects the accuracy of interpretation much less than the prompting style: The error rate for IRSA is on the order of $\sim 10\%$, while for the scratchpad it is $\sim 40\%$. Although the few-shot trace approach is sufficient for simple arithmetic manipulations of variables, it struggles more when it comes to executing control structures, which have multiple reasoning steps: when to test the condition, whether the condition is met, and where to go next to continue the trace. With a simple trace, these three components must be inferred implicitly, but the addition of the IRSA-style annotation explicitly outlines how to approach each of those steps in an auto-regressive fashion rather than making logical leaps.

## A.2 THE LONGEST SUBSTRING WITHOUT REPEATING CHARACTERS

To solve the longest substring without repeating characters problems with basic IRSA, we developed Prompt A.5 based on the 1-index version of the following single-pass algorithm. Interestingly, this algorithm trades computation for memory by creating one variable per unique letter in the sequence for storing the location where the letter was last seen in the sequence during the pass (`last_ind`):

```
# s contains the given string
last_ind = {}
m_len = 0

# window start
st_ind = 0

for i in range(0, len(s)):
  if s[i] in last_ind:
    st_ind=max(st_ind,last_ind[s[i]]+1)

  # Update result if window is longer
  m_len = max(m_len, i-st_ind + 1)

  # Update last index of the character
  last_ind[s[i]] = i
return m_len
```

### A.2.1 BALANCED PARENTHESES

To address the parentheses problem, we used the single execution path that demonstrates stack operations for determining whether the sequence is balanced or not. The beginning and the end are shown in Prompt A.6. For brevity, we have omitted certain portions (represented by ellipses). Note that creating long prompts is made easier by GPT's completion capabilities, i.e., by starting with a description of a few steps and asking the model to finish it. Wherever we want the prompt to differ from the model's guess, we erase the generated text from that point and continue typing our correction/instruction and try to autocomplete again. (See also Sections A.3, A.1 in the Appendix). But interestingly, as discussed in Section 2.2 on fragmented prompting, parts of the execution paths can be omitted: Prompt A.6 as is, with the ellipsis instead of 10 steps in the algorithm, still achieves 91% accuracy!

## A.3 FULL DISCUSSION SECTION

Iteration by Regimenting Self-Attention (IRSA) is a technique for *triggering code execution in GPT-3 models*. Note that the goal is different from the goal of Alphacode Li et al. (2022b) and Copilot Chen et al. (2021); Peng et al. (2023), which are meant to *write* the code, without necessarily understanding what it outputs. While there are indeed examples of rather impressive code generation and even, anecdotally, execution path generation using minimal prompting in the latest Codex and GPT-3 models, the lack of control in current LLMs prevents the consistent achievement of these feats with precision, which is why the code generation applications involve humans in the loop. For instance, as illustrated in zero-shot bubble sort code Prompt A.10, when relying on Codex alone to attempt code execution, the generated samples are intuitively close to the correct solution, but a bit off, preventing correct execution. IRSA, on the other hand, can produce consistently accurate outputs.

In algorithm design, trading computation for memory use is a recurrent idea. IRSA as a technique for LLM inference can be seen in a similar light: We could train a bigger model on more data, with attention spanning deeper into the past tokens, hoping that it could answer a simple yet computationally complex query in just a couple of tokens directly; or we can devise a prompting strategy instructing a smaller LLM to use its token stream as a memory tape, allowing it to reach similar functionality with increased token usage. By triggering and controlling iterative behaviour, we can, in principle, execute arbitrary algorithms, which further raises interesting questions: What are the consequences of LLMs becoming Turing-complete? And how difficult is it to program via IRSA? Will larger GPT models become capable of executing programs correctly without IRSA? Based on our experience in designing the prompts we showed here, we speculate on these three questions in this section.

### A.3.1 POSSIBLE CONSEQUENCES

**(Teaching) Coding.** The integration of LLMs' code generation capabilities with IRSA leads to innovative applications in code generation. Some of it is implied in the interpreter/compiler Prompt A.2, which instructs GPT how to interpret and execute code. Following these ideas, exploring program verification and automatic debugging could be a promising direction. Another obvious application of IRSA is in computer science education, where we often expect students to execute programs on paper to determine what the state will be at some point during the execution. Furthermore, IRSA may also point to new ways of programming by example.

**Adversarial applications.** Any time a computational medium is Turing-complete, a variety of malicious uses may become possible, such as creating and executing malware, exploiting system vulnerabilities, conducting cryptographic attacks, causing resource exhaustion, etc. Thus we should be aware of the double-edged sword with the increased versatility and computational power of GPT models.

**In-context learning and LLM evaluation.** Prompting with IRSA must be considered a zero- or one-shot learning technique, analogous to chain-of-thought prompting. If, via IRSA, LLMs can be disciplined with a regimented prompt to execute arbitrary algorithms involving (double) loops, they may be able to solve arbitrary problems NLP researchers can compose, incorporating natural language understanding and iterative reasoning like belief propagation, constraint satisfaction, search, etc. This renders many of the hard BIG-bench tasks easier than they initially appear, as already suggested by Suzgun et al. (2022) using classical CoT prompting. Many CoT results can be further improved with IRSA (as logical deductions with Prompt 2).

However, triggering such iterative behaviour may still be hampered by the same sensitivity of in-context learning to accidental misleading patterns, already observed in classical prompting (Lu et al., 2022; Zhao et al., 2021), where there may exist a "fantastical" crafting of the prompt that significantly improves the accuracy of the task. In fact, iterative reasoning may further amplify the fantastical choices. Thus, if one LLM successfully solves a hard logical reasoning task using a suitable prompt while another does not, this might imply that the optimal prompt has not yet been found. In fact, it would not be surprising if better prompts are eventually found that enable the LLM we used here (GPT-3, CODE-DAVINCI-002) to solve all tasks with $100\%$ accuracy. Thus, evaluating LLMs on their in-context learning abilities is of questionable value: Some of the hard tasks in BIG-bench may be better suited to evaluating the skills of prompt engineers rather than the LLMs themselves.

**Hybrid models – LLMs as translators.** If LLMs are Turing-complete and can transform problems described in natural language into algorithmically solvable programs, the decision to let them execute the program or not becomes a practical matter of computational cost. With the apparent magic of savant-like guessing gone, it is much more practical to run the algorithms on a classical computer, an approach taken by, for example, Ozturkler et al. (2023) where the external computational mechanism performs probabilistic inference, or Khot et al. (2022) that involves external control flows, and many other recent published and unpublished experiments combining LLMs with external calls and tools Parisi et al. (2022); Gao et al. (2022); Yao et al. (2022); Press et al. (2022); Schick et al. (2023); Paranjape et al. (2023). Such hybrid models could separate the higher level reasoning "System 2" – to use an analogy with models of human cognitive processes (Tversky & Kahneman, 1974; Kahneman, 2011) – from the lower-level "knee-jerk reaction" reasoning "System 1", however savant-like it might be. In such systems, LLMs can dramatically improve traditional artificial intelligence algorithms simply by translating the problems into an appropriate form: see Prompt A.7 where the logical deduction task is solved by creating a call to the `Solve` command in Wolfram

language (Mathematica) for an example. The artificial intelligence community is increasingly interested in researching such systems, e.g., (Bengio, 2017; Goyal & Bengio, 2020), and the developer community is already developing and deploying hybrid language models (Bing-ChatGPT integration, for instance).

**Self-attention control in training and inference.** To paraphrase an old adage on parenting, researchers have spent a lot of effort teaching GPTs to pay attention to everything in the text, and now IRSA is an attempt to stop it from attending to everything. We accomplish it both by drawing attention with a strong repetitive structure and by brute force through skip attention (Section 2.3). More flexible ways of determining what the model should attend to may be needed both in model building and inference.

### A.3.2 PITFALLS OF PROGRAMMING IN GPT-3

Prompts we experimented with induce single loop or double loop program execution. Generally, controlling double loop algorithms, such as Bubble Sort and Longest Common Subsequence, is more challenging. The difficulty lies not in understanding the double loop logic, but rather in the increased probability of running into some of the problems described below. These problems are not always obvious, but can result in a wide range of accuracies achieved by seemingly similar prompts. For example, the two prompt designs for Bubble Sort both worked surprisingly well, but showed a big gap in performance between them (74% and 100%). Here are some tips for attempting IRSA.

**Keep a complete state.** While it is often possible to instruct by analogy without fully accounting for all decisions, keeping the full state (i.e., showing it repeatedly after each transition) is usually preferable. For example, Prompt 3 contains the iterator variable in the state, while Prompt 1 does not. Not only does keeping full state help regiment the attention, but it makes fragmented prompting and skip-to-state attention possible.

**Explain why before the instruction, not after.** LLMs are autoregressive, which makes them easier to prompt in order: from left to right. Thus, instead of instructing with 'We now swap 4 and 2 because 2<4', we instruct with:

```
Because 4<2 is false we swap 4 and 2
```

Then later in generation, e.g., 'Becasue 5<3 is' will trigger generation of token false and it, in turn, will trigger generation of 'we swap', and so on.

To illustrate this numerically, we generated 100 random Python programs with a single if/else statement with a condition of the form (z - y) % 10 + 1 > 5 and a simple command x = 1 or x = 2 in each branch respectively. The dataset contained an equal number of programs where each branch was entered. We tested two prompts that differed only in the order of the answer (x = 1 or 2) and the reasoning. The prompts contained the same 4 examples, two that enter the if branch and two that enter the else branch, in the following format with either answer first:

```
z = 11
y = 27
if (z - y) % 10 + 2 > 8:
    x = 1
else:
    x = 2
Execute:
{x = 2}
because
z - y = 11 - 27 = -16
-16 % 10 = ?
-16 is negative, so the result is 10 - 6
-16 % 10 = 4
4 + 2 = 6
Therefore,
(z - y) % 10 + 2 = 6
6 > 8 is false
```

or reasoning first:

```
z = 11
y = 27
if (z - y) % 10 + 2 > 8:
    x = 1
else:
    x = 2
Execute:
because
z - y = 11 - 27 = -16
-16 % 10 = ?
-16 is negative, so the result is 10 - 6
-16 % 10 = 4
4 + 2 = 6
Therefore,
(z - y) % 10 + 2 = 6
6 > 8 is false
{x = 2}
```

(IRSA recipe requires entering the reasoning first in the prompt). Although the token length and information in the prompts are identical, with just the order being switched, the generation is very different. When the answer is generated first, the model must make a guess ($x = 1$ or $2$) and then rationalize that guess, whereas when the reasoning is generated first, the model has the benefit of that reasoning in generating its answer. On 100 programs using 4-shot prompts, this leads to 52% accuracy for the answer-first prompt and 99% for the reasoning-first prompt. This is crucial for control structures like `if` statements and `while` loops that inform what is executed next through multiple layers of indirection because the prompt needs to instruct the LLM to react at the right time in the right way.

**Empirical entropy in generation can be reduced by IRSA prompting.** GPT models are conditional token samplers, trained to approximate the training data distribution. In generation, the token distribution depends both on the temperature parameter and the previous tokens in the stream. With careful IRSA prompting, we expect the generator to be not only more accurate, but also more certain. This should be characteristic of all autoregressive models, though better ones may be more robust to prompt design pitfalls discussed here.

To test, we further simplified the example above and ran an experiment using both `text-davinci-003` and `gpt-4` with two almost identical 2-shot prompts for evaluating 20 expressions of the form `(a +- b) % 10 + c`.

The two prompts contain identical information, differing only in two characters in total. They both explain the reasoning steps needed to reach the answer. As in the previous ablation, one prompt shows the answer first, followed by an explanation, which then repeats the answer. Concertely, the prompt contains two examples in this form:

```
(12 + 24) % 10 + 1 = 7
12 + 24 = 36
36 % 10 = ?
36 is positive so
36 % 10 = 6
(12 + 24) % 10 + 1 = 7
[DONE]
```

We refer to this prompting style as "guess then rationalize," because when a model is prompted with examples in this form followed by a new problem, it immediately generates the answer as the very first token after =, possibly helped with making that leap by the explanations in the prompt, but before generating its own problem-specific reasoning/explanation, and finally confirming or correcting the answer with its last generated token. As we show below, the model sometimes fails to use detailed explanations to make a correct initial guess and then the confirmation bias often gets the better of the model at the end, even if the generated reasoning should lead to a different conclusion.

The second style of prompting in this ablation differs from the first in just one character per example: Instead of having the correct final answer (here, 7) both at the beginning and at the end, the example contains the question mark at the beginning:

```
(12 + 24) % 10 + 1 = ?
12 + 24 = 36
36 % 10 = ?
36 is positive so
36 % 10 = 6
(12 + 24) % 10 + 1 = 7
[DONE]
```

We refer to such prompting as "reason then answer," because when a model is prompted with examples in this form, it generates the question mark first, instead of an answer, regardless of the problem, then generates the explanation (or reasoning) and then generates the answer without being burdened by a concrete initial guess, which may be incorrect. In other words, the attention is regimented to focus on the reasoning only, developing the answer in a linear fashion.

We tested both styles of prompting with prompts containing the same two worked-out examples, one containing (a + b) and one with (a - b) in the expression, the first example explaining how to evaluate modulo of a positive number and the other of a negative number. (Modulo of a negative number alone often "confuses" GPT models unless there are instructions in the prompt). Each prompt is tested on the same test set of 20 randomly generated expressions. Each test expression is added to the prompt and evaluated 20 times at each of the five different temperatures, allowing us to compute both the average accuracy *and empirical entropy* over the 20 answers (which for both prompting techniques are found as the last token in generation).

As opposed to the previous ablation, where similar expression evaluations were part of making a decision in an `if` block, here we focused on evaluation of the expression alone, and made the reasoning/explanation parts of both prompts shorter by one step: the last addition (6+1=7 in the example above). When a model is following such instructions, it may insert that step in its generation anyhow, or it may follow the recipe and just write the original expression and the answer by performing that last addition "in its head," if it "decides" to do so. Both prompting styles require this mental leap, but it is harder in case of "guess then reason" prompting style. Because of the repetition of the problem at the end, when the last token (the answer) is being generated, the model's attention is drawn to the first token generated right after the expression (the initial guess in "guess then rationalize" or the question mark in "reason then answer"). And the attention is also drawn to the worked out evaluation of the modulo operation, to which a number is to be added to produce the correct answer. In the "reason then answer" case, the attention to the uninformative question mark does not compete with the attention to the reasoning, but in the "guess then rationalize" case it does. Thus, we hypothesized that the entropy of the generations would differ significantly.

Indeed, as we show in Fig A.1, running the same expression 20 times, the entropy increases for the "guess then rationalize" prompt on both GPT3 and GPT4 as the temperature increases, and the accuracy stays low, at less than 40% for GPT3 and just above 60% for GPT4. For the "reason then answer" prompt, both GPT3 and GPT4 have consistently near-zero entropy (sitting at 0 for GPT4) and a near 100% accuracy (sitting at 100% for GPT4). The style of forcing the reasoning before determining an answer sufficiently regiments the attention to reduce the effect of temperature (and potentially other sources of uncertainty due to training data) while still maintaining high accuracy.

As the previous ablation, this experiment illuminates the advantages of IRSA over the original scratchpad prompting that we numerically demonstrated in the main text (Table 3). But, it also further demonstrates how IRSA prompting can create correct long execution traces where any single token could ruin the answer. Even though GPT is a statistical model, where the sampling of each can derail the generation into a "hallucination", regimented attention can dramatically reduce the uncertainty, here created through variation of the temperature, but also inherently present in the model even at zero temperature due to the large training data falling into many different categories and often containing contradictions. The experiment also demonstrates that although most of our experiments are performed with GPT-3 family, the newest (for now) GPT-4 also benefits from IRSA (see also Section A.3.3)

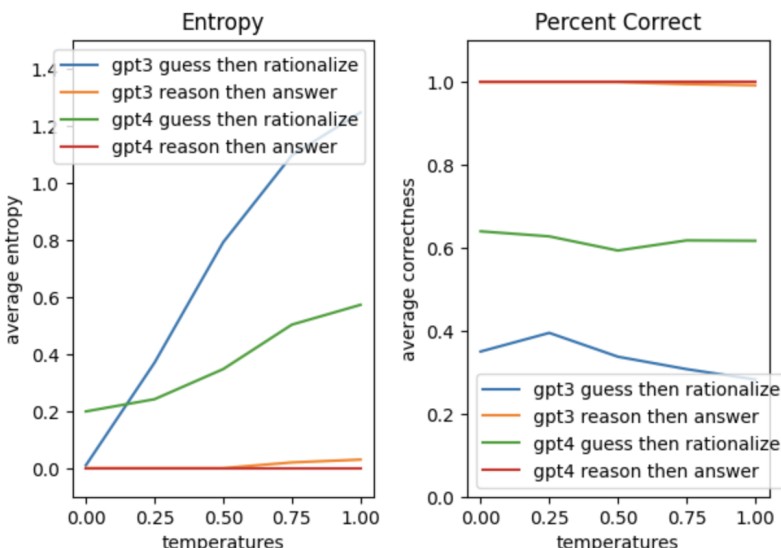

Figure A.1: The average entropy and correctness calculated across 20 trials of 20 expressions on both GPT3 and GPT4 with either a guess first or reason first prompt. We show that the reason first prompt leads to low entropy and high accuracy regardless of model and temperature, while the guess first prompt leads to increased entropy as temperature increases and lower accuracy.

**Avoid unnecessary variation, follow strong structure.** We used the term *regimenting* attention in the naming of the technique to emphasize that strong structure is even more important in IRSA than in other prompting applications. It is usually crucial to order the variables in the state always in the same order, utilize the same keywords to designate the state, use the same language to explain the transitions, and ensure consistent capitalization, punctuation, and even spacing/tabulation. We experimented with several variants of the Bubble Sort prompt, and even when using the same worked-out example, the accuracy can vary dramatically (Although better models are more forgiving than the older ones).

**Generate as much of the prompt with LLM itself.** One way to create such a strong structure is to let the model continue the prompt we are designing after every few lines (going back to correct the incorrectly generated continuation). The model is more likely to stay faithful to the pattern human started than the human is (with spacing, typos, and so on). Because of this, using the interpreter/compiler Prompt A.2 to create an LCS execution path to serve as a prompt is a safer way of generating an IRSA-inducing prompt (as long as we verify that the exemplary execution path is correct).

**Overlapping patterns can be problematic.** When generating the next token, an LLM has to balance many influences of patterns both in the prompt and the so-far generated text. For example, in the LCS algorithm execution Prompt A.3, the model has to balance the long-range self-attention when deciding the next token after `C[1,1]=` with the short-range influences, which make the token `1` most likely after two `1`s in a row regardless of the longer context. At times, short-range influences prevail and cause an incorrect execution. But, long-range self-attention can also inappropriately overrule correct short-range reasoning. For instance, when generating based on the Bubble Sort Prompt 3, the model generates repetitive text that includes many statements of the form 'Because $n<m$ is true/false ...,' which can create strong pattern overruling local evaluation of the next inequality. To demonstrate that, we evaluated the likelihood of the next token after 'Because 2<1 is' for different lengths of context preceding this text. The context had between 1 and 15 lines of text in the form 'Because 2<m is true we ...' with $m \in [3..9]$ randomly chosen, e.g.

```
Because 2<3 is true we ...
```

```
Because 2<7 is true we ...
Because 2<5 is true we ...
Because 2<1 is
```

As we show in Fig A.2, although the preceding context is correct when evaluating the inequalities, the log odds of an incorrect evaluation of `2<1` increase by over six orders of magnitude with the length of this context. The longer this context is, the more it reinforces the pattern 'Because 2< ... true': If 2 was smaller than a variety of numbers, then it is smaller than 1, too! Furthermore, there is a large variation due to the random selection of $m$ in the examples in the context, indicating a variety of other patterns that drive the generation (The figure shows the band between the maximum and minimum log odds over 20 runs). For the contexts of length 7 the odds of picking `true` over `false` become roughly even. IRSA can drive probabilities to be so taut that rerunning the same API call with zero temperature can sometimes return a different result (The code behind the API presumably always adds a very small constant to log probabilities before sampling). Skip-to-state strategy in Section 2.3 is thus less sensitive to patterns that result from program execution.

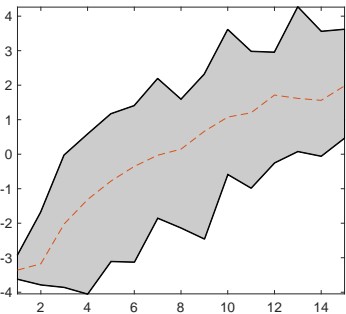

Figure A.2: The difference between GPT Codex log probabilities of tokens `true` and `false` after 'Because 2<1 is', which was preceded by a long context of variable length (x-axis). The context contains between 1 and 15 lines of text comparing number 2 with randomly chosen *larger* numbers and declaring, e.g., `Because 2<6 is true ...` We show the band between the maximum and minimum log odds over 20 trials, as well as the mean of the difference. When the preceding context does not have too many comparisons of 2 with larger numbers, the model overwhelmingly prefers the correct evaluation `false`, but when the context is longer than 7 statements, the model usually prefers `true`.

This fragility further emphasizes the difficulty in evaluating LLMs on in-context learning tasks: Improving accuracy may simply be a matter of spending more time designing a prompt (becoming a GPT whisperer). Still, getting GPT to execute the algorithms studied here was not excessively hard, and it may even become easier on newer models.

### A.3.3 AND WHAT ABOUT GPT-4?

A recent qualitative analysis of GPT-4 abilities Bubeck et al. (2023) includes one example of detailed execution of a Python program for one input (in their Fig. 3.7). The LCS algorithm is well-known, so would the newer and better GPT-4 model execute it correctly and consistently across different inputs? In Prompt A.12, we show a prompt that simply asks GPT-4 to show the LCS algorithm, execute it, and report the result. On our LCS-S dataset, using this prompt design and sampling with zero temperature, GPT-4 gets the correct answer 49% of the times, just slightly better than the 'Guessing' baseline (Table 1). An alternative prompt shown in Prompt A.13, asks for intermediate steps of execution to be shown before the answer is generated, moving the prompting strategy closer to IRSA. This prompt can be thought of as a version of Prompt A.2, but lighter and more straightforward, expecting GPT-4 to be able to show program execution without strict specifications. This prompt leads to the accuracy of 69% on LCS-S, still behind IRSA result with codex (93%, Table 2). To illustrate why this may be, in Prompt A.14 we show the same prompt asking for intermediate steps, but for a different input. The inputs in Prompts A.13 and A.14 were processed differently, even though everything else in the

prompts was the same, and API calls were made with zero temperature. In one case, only the initial and end states of the "dp" matrix are shown, while in the other, several steps (but not all!) are shown. Therefore, it seems that GPT-4 is still hard to control without regimenting self-attention more strictly.

## A.4 FULL SET OF PROMPTS

Here we list the prompts used in all tasks and discussed above. The caption of each prompt contains a saved link to the OpenAI playground.

**Prompt A.1.** Response (shaded green) to Prompt 1, whose ending part is shown.

```
...
        State: a=[1 2 3 5], n_swaps=2, swap_flag=false EndState
        swap_flag is false, so stop the iteration
Final List: 1, 2, 3, 5
Number of swaps: 2
END OF EXECUTION

Problem: 0, 3, 8, 5, 6
EXECUTION
```

```
    Prep
    Length of the list: 5
    Number of consecutive pairs: 4
    a=[0 3 8 5 6]
    set n_swaps=0
    EndPrep
    Iteration:
       set swap_flag=false. The state is:
       State: a=[0 3 8 5 6], n_swaps=0, swap_flag=false EndState
       Pair a[1,2] = [0 3] Check if 0<3. Is it true? Yes.
                           Because of that, we leave state as is
       State: a=[0 3 8 5 6], n_swaps=0, swap_flag=false EndState
       Pair a[2,3] = [3 8] Check if 3<8. Is it true? Yes.
                           Because of that, we leave state as is
       State: a=[0 3 8 5 6], n_swaps=0, swap_flag=false EndState
       Pair a[3,4] = [8 5] Check if 8<5. Is it true? No.
                           Thus, we set swap_flag=true, increase n_swaps by one,
                           and in the latest a=[0 3 8 5 6]
                           swap 8 and 5 to get into state:
       State: a=[0 3 5 8 6], n_swaps=1, swap_flag=true EndState
       Pair a[4,5] = [8 6] Check if 8<6. Is it true? No.
                           Thus, we set swap_flag=true, increase n_swaps by one,
                           and in the latest a=[0 3 5 8 6]
                           swap 8 and 6 to get into state:
       State: a=[0 3 5 6 8], n_swaps=2, swap_flag=true EndState
       swap_flag is true, so do another iteration
    Iteration:
       set swap_flag=false. The state is:
       State: a=[0 3 5 6 8], n_swaps=2, swap_flag=false EndState
       Pair a[1,2] = [0 3] Check if 0<3. Is it true? Yes.
                           Because of that, we leave state as is
       State: a=[0 3 5 6 8], n_swaps=2, swap_flag=false EndState
       Pair a[2,3] = [3 5] Check if 3<5. Is it true? Yes.
                           Because of that, we leave state as is
       State: a=[0 3 5 6 8], n_swaps=2, swap_flag=false EndState
       Pair a[3,4] = [5 6] Check if 5<6. Is it true? Yes.
                           Because of that, we leave state as is
       State: a=[0 3 5 6 8], n_swaps=2, swap_flag=false EndState
       Pair a[4,5] = [6 8] Check if 6<8. Is it true? Yes.
                           Because of that, we leave state as is
       State: a=[0 3 5 6 8], n_swaps=2, swap_flag=false EndState
       swap_flag is false, so stop the iteration
Final List: 0, 3, 5, 6, 8
Number of swaps: 2
END OF EXECUTION
```

**Prompt A.2.** Interpreter/Compiler prompt that can execute LCS algorithm. Playground link (use 0 temperature)

```
Matrix C contains values, e.g:
C[0,0]=1 C[0,1]=6 C[0,2]=11 C[0,3]=16 C[0,4]=21
C[1,0]=2 C[1,1]=7 C[1,2]=12 C[1,3]=17 C[1,4]=22

To query a value:
Show(C[1,2])
<state> C[1,2]=12 </state>

Query an undefined variable
Show(a)
<state> a=0 </state>

To set a value:
C[0,1]:=8
C[0,1] was 6. Now C[0,1]=8.
a:=5
a was 0. Now a=5.

To query multiple variables:
Show(a, C[0:1,0:4])
<state>
a=5
C[0,0]=1 C[0,1]=8 C[0,2]=11 C[0,3]=16 C[0,4]=21
C[1,0]=2 C[1,1]=7 C[1,2]=12 C[1,3]=17 C[1,4]=22
</state>

Program:
N:=1
for i from 0 to N
    C[i,i]=-3
Execute:
i:=0
C[0,0]:=-3
i:=1
C[1,1]:=-3
Finished with i=N, so done with i loop

Show(C)
<state>
C[0,0]=-3 C[0,1]=8 C[0,2]=11 C[0,3]=16 C[0,4]=21
C[1,0]=2 C[1,1]=-3 C[1,2]=12 C[1,3]=17 C[1,4]=22
</state>

Program:
if a<C[0,4]
    C[0,0]:=5
else
    C[0,1]:=2
Execute:
Check if a<C[0,1]?  a is 5 C[0,1] is 8 Is 5<8?...
  ... Yes. C[0,0]:=5
Done

Show(C[0,0])
<state> C[0,0]=5 </state>

Program:
C[0,2]:=detailed_max(C[0,3],C[0,4])
Execute:
C[0,3] is 16, C[0,4] is 21. C[0,2] is the greater of...
  ...them. C[0,2]:=21
Done

LCS:
Input: TA ATA End of input
LCS Prep:
a[1]=T a[2]=A
b[1]=A b[2]=T b[3]=A
M=2 N=3

LCS program:
Show(a,b,M,N)
for i from 1 to M
    for j from 1 to N
        if a[i]==b[j]
            C[i,j]:=C[i-1,j-1]+1
        else
            C[i,j]:=detailed_max(C[i,j-1],C[i-1,j])
        Show(i, j, M, N, C[0:i,0:N])
Show('END')
Execute:
```

**Prompt A.3.** In response to Prompt A.2, the model returns the correct LCS execution path shown here

```
...

Execute:
```

```
<state>
a[1]=T a[2]=A
b[1]=A b[2]=T b[3]=A
M=2 N=3
</state>
i:=1
j:=1
Check if a[1]==b[1]?  a[1] is T b[1] is A Is T==A?...
   ... No. C[1,1]:=detailed_max(C[1,0],C[0,1])
   ... C[1,0] is 0, C[0,1] is 0. C[1,1] is the greater of
   ...them. C[1,1]:=0
<state>
i=1 j=1 M=2 N=3
C[0,0]=0 C[0,1]=0 C[0,2]=0 C[0,3]=0
C[1,0]=0 C[1,1]=0 C[1,2]=0 C[1,3]=0
</state>
j:=2
Check if a[1]==b[2]?  a[1] is T b[2] is T Is T==T?...
   ... Yes. C[1,2]:=C[0,1]+1
   ... C[0,1] is 0. C[1,2]:=1
<state>
i=1 j=2 M=2 N=3
C[0,0]=0 C[0,1]=0 C[0,2]=0 C[0,3]=0
C[1,0]=0 C[1,1]=0 C[1,2]=1 C[1,3]=0
</state>
j:=3
Check if a[1]==b[3]?  a[1] is T b[3] is A Is T==A?...
   ... No. C[1,3]:=detailed_max(C[1,2],C[0,3])
   ... C[1,2] is 1, C[0,3] is 0. C[1,3] is the greater of
   ...them. C[1,3]:=1
<state>
i=1 j=3 M=2 N=3
C[0,0]=0 C[0,1]=0 C[0,2]=0 C[0,3]=0
C[1,0]=0 C[1,1]=0 C[1,2]=1 C[1,3]=1
</state>
i:=2
j:=1
Check if a[2]==b[1]?  a[2] is A b[1] is A Is A==A?...
   ... Yes. C[2,1]:=C[1,0]+1
   ... C[1,0] is 0. C[2,1]:=1
<state>
i=2 j=1 M=2 N=3
C[0,0]=0 C[0,1]=0 C[0,2]=0 C[0,3]=0
C[1,0]=0 C[1,1]=0 C[1,2]=1 C[1,3]=1
C[2,0]=0 C[2,1]=1 C[2,2]=0 C[2,3]=0
</state>
j:=2
Check if a[2]==b[2]?  a[2] is A b[2] is T Is A==T?...
   ... No. C[2,2]:=detailed_max(C[2,1],C[1,2])
   ... C[2,1] is 1, C[1,2] is 1. C[2,2] is the greater of
   ...them. C[2,2]:=1
<state>
i=2 j=2 M=2 N=3
C[0,0]=0 C[0,1]=0 C[0,2]=0 C[0,3]=0
C[1,0]=0 C[1,1]=0 C[1,2]=1 C[1,3]=1
C[2,0]=0 C[2,1]=1 C[2,2]=1 C[2,3]=0
</state>
j:=3
Check if a[2]==b[3]?  a[2] is A b[3] is A Is A==A?...
   ... Yes. C[2,3]:=C[1,2]+1
   ... C[1,2] is 1. C[2,3]:=2
<state>
i=2 j=3 M=2 N=3
C[0,0]=0 C[0,1]=0 C[0,2]=0 C[0,3]=0
C[1,0]=0 C[1,1]=0 C[1,2]=1 C[1,3]=1
C[2,0]=0 C[2,1]=1 C[2,2]=1 C[2,3]=2
</state>
<state>
END
</state>
```

**Prompt A.4.** Full single path execution path prompt using the state description as in the fragmented Prompt 3. Playground link (use with 0 temperature)

```
Problem: 2, 3, 1, 5
EXECUTION
    Length of the list: L=4
    Number of pairs: P=3
    a=[2 3 1 5]
    set n_swaps=0. set i=P=3. set swap_flag=true.
        <state> a=[2 3 1 5] i=3 P=3 n_swaps=0 swap_flag=true </state>
    Since i=3 and P=3, i and P are equal, so this iteration is done, but swap_flag is true,
    so we need another iteration
    Iteration:
        set swap_flag=false.  set i=0. The state is:
        <state> a=[2 3 1 5] i=0 P=3 n_swaps=0 swap_flag=false </state>
        Since i=0 and P=3, these two are different, so we continue
        a[i]=a[0]=2 a[i+1]=a[1]=3
        Because 2<3 is true we keep state as is and move on by increasing i
        <state> a=[2 3 1 5] i=1 P=3 n_swaps=0 swap_flag=false </state>
        Since i=1 and P=3, these two are different, so we continue
        a[i]=a[1]=3 a[i+1]=a[2]=1
        Because 3<1 is false we set swap_flag=true,increase n_swaps by one, and in a=[2 3 1 5] swap 3 and 1,
        and increase i, and keep P as is to get
        <state> a=[2 1 3 5] i=2 P=3 n_swaps=1 swap_flag=true </state>
        Since i=2 and P=3, these two are different, so we continue
        a[i]=a[2]=3 a[i+1]=a[3]=5
        Because 3<5 is true we keep state as is and move on by increasing i
        <state> a=[2 1 3 5] i=3 P=3 n_swaps=1 swap_flag=true </state>
        Since i=3 and P=3, these two are equal, so this iteration is done, but swap_flag is true,
        so we need another iteration
    Iteration:
        set swap_flag=false.  set i=0. The state is:
        <state> a=[2 1 3 5] i=0 P=3 n_swaps=1 swap_flag=false </state>
        Since i=0 and P=3, these two are different, so we continue
        a[i]=a[0]=2 a[i+1]=a[1]=1
        Because 2<1 is false we set swap_flag=true,increase n_swaps by one, and in a=[2 1 3 5] swap 2 and 1,
        and increase i, and keep P as is to get
        <state> a=[1 2 3 5] i=1 P=3 n_swaps=2 swap_flag=true </state>
        Since i=1 and P=3, these two are different, so we continue
        a[i]=a[1]=2 a[i+1]=a[2]=3
        Because 2<3 is true we keep state as is and move on by increasing i
        <state> a=[1 2 3 5] i=2 P=3 n_swaps=2 swap_flag=true </state>
        Since i=2 and P=3, these two are different, so we continue
        a[i]=a[2]=3 a[i+1]=a[3]=5
        Because 3<5 is true we keep state as is and move on by increasing i
        <state> a=[1 2 3 5] i=3 P=3 n_swaps=2 swap_flag=true </state>
        Since i=3 and P=3, these two are equal, so this iteration is done, but swap_flag is true,
        so we need another iteration
    Iteration:
        set swap_flag=false.  set i=0. The state is:
        <state> a=[1 2 3 5] i=0 P=3 n_swaps=2 swap_flag=false </state>
        Since i=0 and P=3, these two are different, so we continue
        a[i]=a[0]=1 a[i+1]=a[1]=2
        Because 1<2 is true we keep state as is and move on by increasing i
        <state> a=[1 2 3 5] i=1 P=3 n_swaps=2 swap_flag=false </state>
        Since i=1 and P=3, these two are different, so we continue
        a[i]=a[1]=2 a[i+1]=a[2]=3
        Because 2<3 is true we keep state as is and move on by increasing i
        <state> a=[1 2 3 5] i=2 P=3 n_swaps=2 swap_flag=false </state>
        Since i=2 and P=3, these two are different, so we continue
        a[i]=a[2]=3 a[i+1]=a[3]=5
        Because 3<5 is true we keep state as is and move on by increasing i
        <state> a=[1 2 3 5] i=3 P=3 n_swaps=2 swap_flag=false </state>
        Since i=3 and P=3, these two are equal, so this iteration is done, but swap_flag is false, so we are done
    Final List: 1, 2, 3, 5
    Number of swaps: 2
    END OF EXECUTION

Problem: 3, 6, 8, 2, 7
EXECUTION
```

**Prompt A.5.** Prompt that triggers execution of the search for the longest substring without repeating characters. Playground link (use 0 temperature)

```
Input: s = c, b, c, a, b, b
START
Unique letters: a, b, c
Define variables last_a=0, last_b=0, last_c=0
Length of sequence s:  L=6
Because L is 6, the needed number of iterations is 6
set st_ind=1
st m_len=0
set i=1
Iteration 1:
    s(1) is c, so use last_c
    last_c is 0, so  nothing to do here.
    max(m_len,  i-st_ind+1) is max(0, 1-1+1) which is...
    ...max(0,1)=1, so we set m_len=1
    since i is 1, and the letter is c, set last_c=1
    increase i by one
    i=2, st_ind=1, m_len=1, last_a=0, last_b=0, last_c=1
End of iteration 1. But we need to do 6 iterations,...
...so we do another one
Iteration 2:
    s(2) is b, so use last_b
    last_b is 0, so  nothing to do here.
    max(m_len,  i-st_ind+1) is max(1, 2-1+1) which is...
    ...max(1, 2)=2, so we set m_len=2
    since i is 2, and the letter is b, set last_b=2
    increase i by one
    i=3, st_ind=1, m_len=2, last_a=0, last_b=2, last_c=1
End of iteration 2.  But we need to do 6 iterations,...
...so we do another one
Iteration 3:
    s(3) is c, so use last_c
    last_c is greater than 0, so we reason...
    ...max(st_ind, last_c+1) is max(1, 2)=2...
    ...so we set st_ind=2
    max(m_len,  i-st_ind+1) is max(2, 3-2+1) which is...
    ...max(2, 2)=2, so we set m_len=2
    since i is 3, and the letter s(3) is c, set last_c=3
    increase i by one
    i=4, st_ind=2, m_len=2, last_a=0, last_b=2, last_c=3
End of iteration 2. But we need to do 6 iterations,...
...so we do another one
Iteration 4:
    s(4) is a, so use last_a
    last_a is 0, so  nothing to do here.
    max(m_len,  i-st_ind+1) is max(2, 4-2+1) which is...
    ...max(2, 3)=3, so we set m_len=3
    since i is 4, and the letter s(4) is a, set last_a=4
    increase i by one
    i=5, st_ind=2, m_len=3, last_a=4, last_b=2, last_c=3
End of iteration 4. But we need to do 6 iterations,...
...so we do another one
Iteration 5:
    s(5) is b, so use last_b
    last_b is greater than 0, so we reason...
    ...max(st_ind, last_b+1) is max(2, 2+1) which is...
    ...max(2, 3)=3 so we set st_ind=3
    max(m_len,  i-st_ind+1) is max(3, 5-3+1) which is...
    ...max(3, 3)=3, so we set m_len=3
    since i is 5, and the letter s(5) is b, set last_b=5
    increase i by one
    i=6, st_ind=3, m_len=3, last_a=4, last_b=5, last_c=3
End of iteration 5. But we need to do 6 iterations,...
...so we do another one
Iteration 6:
    s(6) is b, so use last_b
    last_b is greater than 0, so we reason...
    ...max(st_ind, last_b+1) is max(3, 5+1) which is...
    ...max(3, 6)=6 so we set st_ind=6
    max(m_len,  i-st_ind+1) is max(3, 6-6+1) which is...
    ...max(3, 1)=3, so we set m_len=3
    since i is 6, and the letter s(6) is b, set last_b=6
    increase i by one
    i=7, st_ind=6, m_len=3, last_a=4, last_b=6, last_c=3
End of iteration 6. We needed to do 6 iterations,...
...so we are done

The solution is: m_len=3
END

Input: s = p, w, w, k, e, p, z
START
```

**Prompt A.6.** Prompt that triggers evaluation of parentheses using a stack. Full prompt in playground, and Prompt as here (with 10 steps missing) in playground. Meant to be used with 0 temperature.

```
input: ) [ { } ] ( { } ) [ ( { } ) ] } {
input wriritten as a sequence of symbols:
s= ')', '[', '{', '}', ']', '(', '{','}', ')', '[',
'(', '{', '}', ')',']', '}', '{'
length(s)= 17
stack is initialized as empty
i=0
there is nothing in stack, so push s(0)= ')' on stack
stack= )
are the last two symbols an open and a closed
parenthesis of the same type? No. Stack stays same.
i=1
we push s(1)='[' on the stack
stack= ) [
are the last two symbols an open and a closed
parenthesis of the same type? No. Stack stays same.
i=2
we push s(2)='{' to the stack
stack= ) [ {
are the last two symbols an open and a closed
parenthesis of the same type? No. Stack stays same.
i=3
we push s(3)='}' to the stack
stack= ) [ { }
are the last two symbols an open and a closed
parenthesis of the same type? Yes, they are { },
opening then closing.
We pop the last two symbols from the stack.
stack= ) [
i=4
we push s(4)=']' to the stack
stack= ) [ ]
are the last two symbols an open and a closed
parenthesis of the same type? Yes, they are [ ],
opening then closing.
We pop the last two symbols from the stack
stack= )
i=5
we push s(5)='(' to the stack
stack= ) (

...

i=15
we push s(15)='}' to the stack
stack= ) }
are the last two symbols an open and a closed
parenthesis of the same type? No. Stack stays same.
i=16
we push s(16)='{' to the stack
stack= ) } {
are the last two symbols an open and a closed
parenthesis of the same type? No. Stack stays same.
i=17
we have reached the end of the input string.
If the stack has some parenthesis left in it,
the sequence is invalid, otherwise,
if the stack is empty, it is valid.
Sequence is: invalid
END

input:
```

**Prompt A.7.** A prompt (white background) for translating logical deduction puzzles from BIG-bench into a Wolfram language (Mathematica) call. The response from GPT-3 Codex is shown green-shaded. Playground link (use 0 temperature)

```
Puzzle: The following objects need to be ordered. obj1 is the biggest. obj2 is smaller than obj3.
obj1 is bigger than obj2.

Question: Which object is the biggest?

Parsing step:
    Items: obj1, obj2, obj3
    Numbe of items: 3
    Statement 1: obj1 is the biggest.
    Statement 2: obj2 is smaller than obj3.
    Statement 3: obj1 is bigger than obj2.

Scoring identification step:
     Scores will refer to size.
     Since we have 3 items, let's assume that the biggest gets a score  of 3 pounds and the smallest gets
     the score of 1 pound.

Translation step:
    Available variable names: x, y, z, a, b, c
    Map item scores of 'obj1', 'obj2', 'obj3' to variable names x, y, z
    obj1 score is x; obj2 score is y; obj3 is z;
    Statement 1: 'x' is the biggest.
    Statement 2: 'y' is smaller than 'z'.
    Statement 3:  'x' is bigger than 'y'.
    Domain constraints: 1<=x<=3, 1<=y<=3, 1<=z<=3
    Statemets as a set of inequalities or equalities: x==3, y<z, x>y
    All varaibles are different: x!=y, x!=z, y!=z

Call to WolframAlpha:
    Solve[{1<=x<=3, 1<=y<=3, 1<=z<=3, x==3, y<z, x>y, x!=y, x!=z, y!=z }, {x, y, z}, Integers]

END

Puzzle: A fruit stand sells five fruits: kiwis, pears, peaches, loquats, and apples.
The loquats are more expensive than the kiwis. The pears are more expensive than the peaches.
The apples are more expensive than the loquats. The apples are less expensive than the peaches.

Question: Which is the most expensive?
```

```
Parsing step:
    Items: kiwis, pears, peaches, loquats, apples
    Numbe of items: 5
    Statement 1: The loquats are more expensive than the kiwis.
    Statement 2: The pears are more expensive than the peaches.
    Statement 3: The apples are more expensive than the loquats.
    Statement 4: The apples are less expensive than the peaches.

Scoring identification step:
     Scores will refer to price.
     Since we have 5 items, let's assume that the most expensive gets a score  of 5 dollars
     and the least expensive gets the score of 1 dollar.

Translation step:
    Available variable names: x, y, z, a, b, c
    Map item scores of 'kiwis', 'pears', 'peaches', 'loquats', 'apples' to variable names x, y, z, a, b
    kiwis score is x; pears score is y; peaches is z; loquats is a; apples is b;
    Statement 1: 'a' is more expensive than 'x'.
    Statement 2: 'y' is more expensive than 'z'.
    Statement 3:  'b' is more expensive than 'a'.
    Statement 4:  'b' is less expensive than 'z'.
    Domain constraints: 1<=x<=5, 1<=y<=5, 1<=z<=5, 1<=a<=5, 1<=b<=5
    Statemets as a set of inequalities or equalities: a>x, y>z, b>a, b<z
    All varaibles are different: x!=y, x!=z, x!=a, x!=b, y!=z, y!=a, y!=b, z!=a, z!=b, a!=b

Call to WolframAlpha:
    Solve[{1<=x<=5, 1<=y<=5, 1<=z<=5, 1<=a<=5, 1<=b<=5, a>x, y>z, b>a, b<z, x!=y, x!=z, x!=a, x!=b,
    y!=z, y!=a, y!=b, z!=a, z!=b, a!=b}, {x, y, z, a, b}, Integers]
```

**Prompt A.8.** The Nye et al. (2021) execution trace prompt for interpreting synthetic Python programs. For two- and three-shot traces, the additional examples were added in the gray shaded area. 1-shot Playground link, 2-shot Playground link, 3-shot Playground link (use 0 temperature)

```
Consider the following Python function:

def f(v0):
    v0 += 0
    v4 = 2
    while v4 > 0:
        v4 -= 1
        v0 *= 2
    return v0

output = f(6)

What is the execution trace?

[BEGIN]

state: {}
line: def f(v0):
state: {"f": "<callable_object f>"}
line: output = f(6)
state: {"v0": 6}
line:     v0 += 0
state: {"v0": 6}
line:     v4 = 2
state: {"v0": 6, "v4": 2}
line:     while v4 > 0:
state: {"v0": 6, "v4": 2}
line:         v4 -= 1
state: {"v0": 6, "v4": 1}
line:         v0 *= 2
state: {"v0": 12, "v4": 1}
line:     while v4 > 0:
state: {"v0": 12, "v4": 1}
line:         v4 -= 1
state: {"v0": 12, "v4": 0}
line:         v0 *= 2
state: {"v0": 24, "v4": 0}
line:     while v4 > 0:
state: {"v0": 24, "v4": 0}
line:     return v0
state: {"f": "<callable_object f>", "output": 24}

[DONE]
```

```
...
```

```
Consider the following Python function:
```

**Prompt A.9.** The execution trace prompts with IRSA-style explanations added in between line and state for interpreting synthetic Python programs. For two- and three-shot traces, the additional examples were added in the gray shaded area. 1-shot Playground link, 2-shot Playground link, 3-shot Playground link (use 0 temperature)

```
Consider the following Python function:

def f(v0):
    v0 += 0
    v4 = 2
    while v4 > 0:
        v4 -= 1
        v0 *= 2
    return v0

output = f(6)

What is the execution trace?

[BEGIN]

state: {}
line: def f(v0):
explanation: definition of function f that takes an initial value for the variable v0
state: {"f": "<callable_object f>"}
line: output = f(6)
explanation: execute the code under the definition of function f and initialize variable v0 with initial value 6
state: {"v0": 6}
line:     v0 += 0
explanation: v0 += 0 means v0 = v0 + 0. v0 is 6, so v0 + 0 = 6 + 0 = 6. Now v0 is 6
state: {"v0": 6}
line:     v4 = 2
explanation: initialize variable v4 with initial value 2
state: {"v0": 6, "v4": 2}
line:     while v4 > 0:
explanation: Check v4 > 0. v4 was 2. Is 2 > 0? Yes, so we enter the while loop
state: {"v0": 6, "v4": 2}
line:         v4 -= 1
explanation: v4 -= 1 means v4 = v4 - 1. v4 is 2, so v4 - 1 = 2 - 1 = 1. Now v4 is 1
state: {"v0": 6, "v4": 1}
line:         v0 *= 2
explanation: v0 *= 2 means v0 = v0 * 2. v0 is 6, so v0 * 2 = 6 * 2 = 12. Now v0 is 12
state: {"v0": 12, "v4": 1}
line:     while v4 > 0:
explanation: Check v4 > 0. v4 was 1. Is 1 > 0? Yes, so we enter the while loop
state: {"v0": 12, "v4": 1}
line:         v4 -= 1
explanation: v4 -= 1 means v4 = v4 - 1. v4 is 1, so v4 - 1 = 1 - 1 = 0. Now v4 is 0
state: {"v0": 12, "v4": 0}
line:         v0 *= 2
explanation: v0 *= 2 means v0 = v0 * 2. v0 is 12, so v0 * 2 = 12 * 2 = 24. Now v0 is 24
state: {"v0": 24, "v4": 0}
line:     while v4 > 0:
explanation: Check v4 > 0. v4 was 0. Is 0 > 0? No, so we are done with the loop
state: {"v0": 24, "v4": 0}
line:     return v0
explanation: Return the current value of v0, which is 24, to replace the initial function call
state: {"f": "<callable_object f>", "output": 24}

[DONE]

...

Consider the following Python function:
```

**Prompt A.10.** A zero-shot code prompt (white background) which, when temperature is carefully chosen may trigger code execution (shaded green) by Codex. Although the sequence gets sorted, the execution goes awry after that. Importantly, the number of swaps is not tracked, even though the code contains appropriate instructions. In contrast, IRSA prompts consistently and repeatedly trigger correct program execution. See also Prompts A.13 and A.14 for examples of inconsistent behaviour of GPT-4 with zero-shot code prompting.

```
# Here is the code for Bubble Sort algorithm that keeps track of the number of needed swaps:
def bubbleSort(arr):
    n = len(arr)
    # initiate the number of swaps to 0
    n_swaps=0;
    # Traverse through all array elements
    for i in range(n):

        # Last i elements are already in place
        for j in range(0, n-i-1):

            # traverse the array from 0 to n-i-1
            # Swap if the element found is greater
            # than the next element
            if arr[j] > arr[j+1] :
                arr[j], arr[j+1] = arr[j+1], arr[j]
                n_swaps = n_swaps+1

Problem: 8, 7, 6, 4, 9
EXECUTION:
```

```
    i=0
    j=0
    8 > 7
    7, 8, 6, 4, 9
    j=1
    8 > 6
    7, 6, 8, 4, 9
    j=2
    8 > 4
    7, 6, 4, 8, 9
    j=3
    8 > 9
    7, 6, 4, 8, 9
    j=4
    i=1
    j=0
    7 > 6
    6, 7, 4, 8, 9
    j=1
    7 > 4
    6, 4, 7, 8, 9
    j=2
    7 > 8
    6, 4, 7, 8, 9
    j=3
    i=2
    j=0
    6 > 4
    4, 6, 7, 8, 9
    j=1
    6 > 7
    4, 6, 7, 8, 9
    j=2
    i=3
    j=0
    4 > 6
    4, 6, 7, 8, 9
    j=1
    i=4
    j=0
    i=5
    j=0
    i=6
    j=0
    i=7
    j=0
    i=8
    ...
```

**Prompt A.11.** An example of a baseline prompt design. Prompts with and without the code, shaded in gray, were used. Zero shot prompts omitted the examples, also shaded in gray. Other than the (buggy) algorithm we adopted for the logical deduction puzzle, the algorithms we tested are known to Codex, and we based the codes used in the prompts on the ones generated by Codex itself.

```
# Python3 program to find the length
# of the longest substring
# without repeating characters in string s
# the maximum length of such a substring will be returned in m_len
```

```
def longestUniqueSubsttr(s):
    # last index of every character
    last_idx = {}
    m_len = 0
    # starting index of current
    # window to calculate m_len
    start_idx = 0
    for i in range(0, len(s)):
        # Find the last index of str[i]
        # Update start_idx (starting index of current window)
        # as maximum of current value of start_idx and last
        # index plus 1
        if s[i] in last_idx:
            start_idx = max(start_idx, last_idx[s[i]] + 1)

        # Update result if we get a larger window
        m_len = max(m_len, i-start_idx + 1)
        # Update last index of current char.
        last_idx[s[i]] = i
    return m_len
```

```
What would the algorithm above, which looks for the longest substring with non repeating characters
compute for a given problem? Use this format:

Input: s = ...
START
The solution is: m_len= ...
END
```

```
Input: s = a, a, c, a, d, b, b
START
The solution is: m_len=4
END

Input: s = b, b, a, a, d, b, b
START
The solution is: m_len=3
END

Input: s = c, a, d, c, b, c, a
START
The solution is: m_len=4
END

Input: s = a, c, b, a, c, d, d
START
The solution is: m_len=4
END

Input: s = b, b, d, d, a, d, b
START
The solution is: m_len=3
END
```

```
Input: s = a, c, d, c, a, c, a
START
```

**Prompt A.12.** Given the non-shaded text as a prompt, GPT-4 (0 temperature) returns the green shaded response.

```
We need to compute the longest common subsequence for two sequences

s1=bccba
s2=ccaa

using the dynamic programming algorithm. Show the python code for the algorithm, and then execute it.
Finally, output the length of the longest common subsequence bracketed with <answer> and </answer>.
```

```
Here's the Python code for the dynamic programming algorithm to find the longest common subsequence:

'''python
def longest_common_subsequence(s1, s2):
    m = len(s1)
    n = len(s2)
    dp = [[0] * (n + 1) for _ in range(m + 1)]

    for i in range(1, m + 1):
        for j in range(1, n + 1):
            if s1[i - 1] == s2[j - 1]:
                dp[i][j] = dp[i - 1][j - 1] + 1
            else:
                dp[i][j] = max(dp[i - 1][j], dp[i][j - 1])

    return dp[m][n]

s1 = "bccba"
s2 = "ccaa"
lcs_length = longest_common_subsequence(s1, s2)
print(f"<answer>{lcs_length}</answer>")
'''

Now, let us execute the code and find the length of the longest common subsequence:

'''python
s1 = "bccba"
s2 = "ccaa"
lcs_length = longest_common_subsequence(s1, s2)
print(f"<answer>{lcs_length}</answer>")
'''

Output:

'''
<answer>3</answer>
'''
```

**Prompt A.13.** Given the non-shaded text as a prompt, GPT-4 (0 temperature) returns the green shaded response. The prompt attempts to trigger tracking steps of program execution, but only the initial and final matrix are shown.

```
We need to compute the longest common subsequence for two sequences

s1=bccba
s2=ccaa

using the dynamic programming algorithm. Show the python code for the algorithm, and then write down its
execution with intermediate steps. Finally, output the length of the longest common subsequence bracketed
with <answer> and </answer>.
```

```
Here's the Python code for the dynamic programming algorithm to find the longest common subsequence:

'''python
def longest_common_subsequence(s1, s2):
    m = len(s1)
    n = len(s2)
    dp = [[0] * (n + 1) for _ in range(m + 1)]

    for i in range(1, m + 1):
        for j in range(1, n + 1):
            if s1[i - 1] == s2[j - 1]:
                dp[i][j] = dp[i - 1][j - 1] + 1
            else:
                dp[i][j] = max(dp[i - 1][j], dp[i][j - 1])

    return dp[m][n]

s1 = "bccba"
s2 = "ccaa"
print(longest_common_subsequence(s1, s2))
'''

Now let us s go through the execution with intermediate steps:

1. Initialize the 'dp' table with dimensions (m+1) x (n+1), where m and n are the lengths of
s1 and s2, respectively. The table will look like this:

    0 0 0 0 0
    0 0 0 0 0
    0 0 0 0 0
    0 0 0 0 0
    0 0 0 0 0
    0 0 0 0 0

2. Iterate through the table, comparing characters from s1 and s2. Update the table based
on the dynamic programming formula:

    0 0 0 0 0
    0 0 1 1 1
    0 1 1 1 1
    0 1 2 2 2
    0 1 2 2 2
    0 1 2 2 3

3. The length of the longest common subsequence is in the bottom-right cell of the table, which is 3.

So, the length of the longest common subsequence is <answer>3</answer>.
```

**Prompt A.14.** Given the same non-shaded text as in Prompt A.13 as a prompt, **just for a different input**, GPT-4 (0 temperature) returns the green shaded response that this time does list more (but not all!) intermediate steps. This style of prompting GPT-4 reaches accuracy of 69% on LCS-S, compared to 93% achieved with IRSA in GPT-3.

```
We need to compute the longest common subsequence for two sequences

s1=aaca
s2=abab

using the dynamic programming algorithm. Show the python code for the algorithm, and then write down its
execution with intermediate steps. Finally, output the length of the longest common subsequence bracketed
with <answer> and </answer>.
```

```
Here is the Python code for the dynamic programming algorithm to compute the longest common subsequence:
```python
def longest_common_subsequence(s1, s2):
    m = len(s1)
    n = len(s2)

    dp = [[0]*(n+1) for _ in range(m+1)]

    for i in range(m+1):
        for j in range(n+1):
            if i == 0 or j == 0:
                dp[i][j] = 0
            elif s1[i-1] == s2[j-1]:
                dp[i][j] = dp[i-1][j-1] + 1
            else:
                dp[i][j] = max(dp[i-1][j], dp[i][j-1])

    return dp[m][n]

s1 = "aaca"
s2 = "abab"
result = longest_common_subsequence(s1, s2)
print(result)
```
Here is the execution with intermediate steps:

1. Initialize the 'dp' table with dimensions (m+1) x (n+1). For the given sequences, it will look like:

    0 0 0 0 0
0   0 0 0 0 0
0   0 0 0 0 0
0   0 0 0 0 0
0   0 0 0 0 0

2. Fill in the 'dp' table by iterating through each cell and apply the algorithm rules:

For i, j = 1, 1: s1[i-1] != s2[j-1], so dp[i][j] = max(dp[i-1][j], dp[i][j-1]) = max(0, 0) = 0
    0 0 0 0 0
0   0 0 0 0 0
0   0 0 0 0 0
0   0 0 0 0 0
0   0 0 0 0 0

For i, j = 1, 2: s1[i-1] != s2[j-1], so dp[i][j] = max(dp[i-1][j], dp[i][j-1]) = max(0, 0) = 0
    0 0 0 0 0
0   0 0 0 0 0
0   0 0 0 0 0
0   0 0 0 0 0
0   0 0 0 0 0

For i, j = 1, 3: s1[i-1] == s2[j-1], so dp[i][j] = dp[i-1][j-1] + 1 = 0 + 1 = 1
    0 0 0 0 0
0   0 0 0 0 0
0   0 0 1 0 0
0   0 0 0 0 0
0   0 0 0 0 0

Continue filling in the table with the same rules:

    0 0 0 0 0
0   0 0 0 0 0
0   0 0 1 0 0
0   0 1 1 1 0
0   0 1 1 2 0

3. The value in the bottom-right cell contains the length of the longest common subsequence:

<answer>2</answer>
```

