# OpenReview forum: "GPT Is Becoming a Turing Machine: Here Are Some Ways to Program It"
_ICLR.cc/2024/Conference — Submitted to ICLR 2024_

### Official Review · Reviewer_FzpB · 2023-10-28

**Soundness:** 2 fair
**Presentation:** 2 fair
**Contribution:** 3 good
**Rating:** 6
**Confidence:** 3

**Summary:**

In this paper, the author(s) explore the description and execution of iterative algorithms in large language models. In particular, the author(s) propose “iterations by regimenting self-attention” (IRSA) in which they provide a repetitive and comprehensive example in prompts. Moreover, they illustrate two variants, fragmented prompting and skip attention, which can improve accuracy and address token limitations. Further, the author(s) design a “GPT compiler” that can generate execution paths for large language models similar to IRSA.

**Strengths:**

Importance of contribution: The proposed solution can achieve outperformance than state-of-the-art approaches. Meanwhile, it also highlights the significance of prompting engineering as GPT-3 applied IRSA can generate more accurate results than GPT-4 without IRSA.

Soundness: The author(s) explain the approach in detail, and conduct evaluation via comparative analysis regarding different questions.

Quality of presentation: The paper is well-organized, and the language is technical yet understandable for readers with domain knowledge.

Comparison with related works: The author(s) introduce extant studies on large language model prompting.

**Weaknesses:**

- The methodology can be elaborated for better clarity.
- The overall structure of this paper can be adjusted.
- The research gaps can be further highlighted and discussed.

**Questions:**

- Section 2.1: “a variant of Bubble Sort algorithm adapted to this problem and shown in Prompt 2 can be used to solve 76% of these puzzles”, the author(s) should provide evaluation results to support this statement.
- I wonder whether the author(s) consider applying IRSA to GPT-4 and test the result accuracy.
- The author(s) generate random non-repeating digit sequences in the Bubble Sort problem, I wonder whether repeating digits can affect the results.
- I understand the conference has page limits, the authors can consider adjusting the font of prompts to save more space to elaborate the solution description, especially Section 2.4. The font of Prompt 1 is larger than the following prompts.
- The authors should clarify the evaluation metric is accuracy in the caption of Table 1-3.
- The author(s) can consider comprehensively comparing the proposed models with related work to clearly identify the research gap.

---

> ### Author Response · Authors · 2023-11-23
>
> The section 2.1 note on adapting the Bubble Sort algorithm to solve logical puzzles describes that Prompt 2, which was used in evaluating the Logical deduction task, solves 76% of the logical puzzles, and this result is shown in Table 1 in the Logical deduction task under IRSA.
>
> We have applied IRSA to GPT-4 in a couple instances, described in Section A.3.2 and A.3.3. In section A.3.3, we discuss that GPT-4 is not better than GPT-3 with IRSA on the LCS task. In Section A.3.2, the newly added experiment evaluating expressions of the form (a +- b) % 10 + c, averaging entropy and correctness across many trials to compare two different prompts on both GPT-3 and GPT-4, as we increase the temperature, providing a more direct comparison using GPT-4 with IRSA. The first prompt is a “guess then check” style prompt, where the answer to the expression is given first, followed by a step-by-step reasoning to achieve the answer, and the second prompt is a “reason then answer” style prompt, where the first line has a question mark instead of the answer, followed by the same step-by-step reasoning.
>
> The guess then check prompt has two examples of the following form:
>
> (24-41)%10+2=5
>
> 24-41=-17
>
> -17%10=?
>
> -17 is negative so the answer is 10-7=3
>
> -17%10=3
>
> (24-41)%10+2=5
>
> (The additional example has a positive result in a+-b),
> And the “reason than answer” style that IRSA would follow, differs only in replacing the first 5 with a question mark.
>
> Figure A.1 illustrates how the “guess then check” style prompt has just over 60% accuracy with GPT-4 and just less than 40% accuracy with GPT-3, while the IRSA style prompt has 100% accuracy for GPT-4 and near 100% accuracy for GPT-3. Interestingly, comparing the average entropy across 20 trials of 20 expressions shows that as the temperature increases, the entropy increases for both GPT-3 and GPT-4 using the “guess then check” prompt but with the “reason then answer” prompt, the entropy sits at zero for GPT-4 and near zero for GPT-3 regardless of temperature.
>
> On the repeating digits front, it seems that adjusting the prompt (A.4) just slightly to use <= as the comparison rather than < enables GPT-3 and GPT-4 to correctly execute bubble sort with repeating digits. We did a cursory trial with [3, 6, 8, 8, 7], [8, 8, 8, 8, 8], and [8, 8, 8, 8, 7] and there is no issue with execution. Importantly, though it did require the adjustment to the prompt with <= so that the swap_flag was not set to true for switching repeating digits, leading to an infinite loop. Trying with [3, 8, 6, 8, 7] led to an interesting issue with GPT-3 where once the execution got to <state> a=[3 6 8 8 7] i=3 P=4 n_swaps=1 swap_flag=true </state>, it proceeded to correctly identify the last 8 and 7 to swap, but instead created the state <state> a=[3 6 8 7 7] i=4 P=4 n_swaps=2 swap_flag=true </state>, where it ended up repeating the 7 instead of shaping the 7 and 8. Using skip attention, rerunning the prompt but adding the last correct state right after EXECUTION (not allowing GPT-3 to attend to how it generated that state), fixed the problem. GPT-4 did not run into this issue with the same problem.
>
> Thank you for the notes about font size, clarifying evaluation metrics, and more clearly identifying the research gap. We will address those notes as well as elaborate on methodology in our final version.

---

### Official Review · Reviewer_uwZf · 2023-10-28

**Soundness:** 2 fair
**Presentation:** 1 poor
**Contribution:** 2 fair
**Rating:** 3
**Confidence:** 4

**Summary:**

This work proposes IRSA (iterations by regimenting self-attention) for the use of GPT3 model to execute programs that have loops. Its core idea is using an example input and its execution path for constructing highly structured prompts. It further explores combining multiple fragments of execution paths, instead of a prompt that covers entire execution path of any single example. It then skips parts of generated text when performing self-attention for more efficient token use. GPT3+IRSA outperforms GPT4 for executing the programs given the evaluated programs.

**Strengths:**

This is an interesting work.
The proposed approach seems to work given the evaluated programs.

**Weaknesses:**

The proposed work may point out an interesting direction for using LLMs to execute programs, but its current form and results are premature and it’s not ready to be published.

The presentation of IRSA has been majorly illustrated by examples. While such examples are useful, there still lacks a formulation of IRSA. The IRSA prompting for these examples look ad-hoc, and it is not clear how IRSA can be automatically applied to execute general programs, without significant manual efforts.

The example and its execution path play the critical role in IRSA. Isn’t the availability of an execution path too strong assumption for enabling GPT to execute a program? There are many things not discussed, including how to select the example and how to achieve the execution path. Is there one example or multiple examples used in IRSA?

It is not clear why IRSA is not used together with GPT4, which casts doubts on the applicability of IRSA approach.

The evaluation is neither comprehensive nor systematic. The example programs in the evaluation look simplistic.

**Questions:**

1. Is there an algorithmic formulation of IRSA? For example, interested people can refer to this formulation to implement IRSA and execute programs using LLMs.

2. Is the execution path assumption in IRSA too strong? How many examples are used in IRSA?

3. Can IRSA be used with LLMs other than GPT-3?

---

> ### Author Response · Authors · 2023-11-23
>
> The example programs are meant to serve as a more direct comparison against the experiments proposed by Nye et al’s with a trace prompt. Exactly because these programs are simplistic, one might assume the LLM would not have an issue with correct execution, however this is not the case without triggering the model to unroll its reasoning and not only the execution trace. A mere execution trace leads to 54-59% accuracy depending on how many examples are given while adding IRSA style explanations between states improves the accuracy, getting up to 85-91%. In terms of prompts for more complex programs, such as those used for bubble sort, LCS, etc., there need not be an already available execution path if the model can create one itself.
>
> The LLM itself can be leveraged to create IRSA style prompts, as we did to create the LCS execution path using the interpreter/compiler Prompt A.2. Otherwise, it would also be feasible to create an execution trace of a program that adds in an explanation between states for each step of the program. The most important features of IRSA are to describe the conditions for continuing and stopping execution, including explanations for state changes in the form “why then what”, and including full state descriptions. Though as skip attention and fragmented prompting shows, these state descriptions need not amount to a full execution path either — they need only describe the possible state transitions.
>
> Our experiments have a variation in number of examples used. With bubble sort, LCS, logical puzzles, and the parentheses tasks shown in Table 1, there are 1-shot prompts. The fragmented prompts on the bubble sort execution have multiple fragments shown in Table 2 (single path and then 7, 13, 19, and 25 fragments), and the ablations in A.3.2 include 4-shot prompts for the if-else experiment and 2-shot prompts for the newly added expressions experiment (showing entropy and correctness as temperature increases for GPT-3 and GPT-4). Table 3 shows the interpreter prompts for synthetic programs with 1-shot, 2-shot, and 3-shot prompts.
>
> IRSA can certainly be used with LLMs other than GPT-3, and we provide comparison against GPT-4 in a now added experiment on evaluating expressions in section A.3.2 in the appendix. Fig. A.1 studies evaluation of expressions of the form (a +- b) % 10 + c, which are hard due to the potential of a negative mod computation and the need to combine multiple computations. But prompting with worked-out examples may not work perfectly without following the IRSA recipe. We look at the difference between prompting in “guess then check” style,
>
> (24-41)%10+2=5
>
> 24-41=-17
>
> -17%10=?
>
> -17 is negative so the answer is 10-7=3
>
> -17%10=3
>
> (24-41)%10+2=5
>
> (plus an additional example with a positive result in a+-b),
> vs the “reason than answer” style that IRSA would follow, differing only in replacing the first 5 with a question mark.
>
> The only difference is one character per example (5 vs ? in the first line), but as the fig shows, the performance is very different (evaluating 20 randomly generated expressions in 20 trials each). One might assume that with the first prompt, when working on a new expression, the model should guess correctly following the examples, and then even if wrong, it would have a chance of correcting itself at the end, as its generated explanation may change its mind. But, the model often does not guess correctly as the reasoning requires iterative attention that it cannot do without unrolling the reasoning step by step. Then, even after generating a correct follow up step-by-step reasoning and computing correctly (a+b)%10, it has trouble deciding if it should just add c, or follow its initial guess. The second prompt does not create this uncertainty as the model follows the recipe and generates a placeholder ? instead of an initial guess, which later does not conflict with the generated reasoning. The answer entropy quickly rises with temperature (even for GPT-4) for the first prompt, but not for the second. Furthermore, the reduced entropy of the “reason then answer” prompt does not compromise accuracy, with the accuracy of both GPT-3 and GPT-4 at near 100% (sitting at 100% for GPT-4) regardless of temperature, while the “guess then check” style has less than 40% accuracy for GPT-3 and just over 60% accuracy for GPT-4.
>
> A cursory trial of the bubble sort prompts with GPT-4 suggest that the IRSA style prompts triggers the same iterative behavior by GPT-4 as well for more complex algorithms.

---

> > ### Comment · Reviewer_uwZf · 2023-11-23
> >
> > Thank you for the response. I'll not change my score. While this work comprises many useful illustrations, the formalism of an approach is essential and needed.

---

### Official Review · Reviewer_yv3R · 2023-11-01

**Soundness:** 2 fair
**Presentation:** 2 fair
**Contribution:** 2 fair
**Rating:** 5
**Confidence:** 3

**Summary:**

The proposed method essentially gives a full stack trace of a programmatic execution on a problem as a prompt to the LLM, and ask the LLM to solve a task by imitating the content of the stack trace. Thus, the LLM can copy the overall structure of the trace, retaining the rigid programmatic execution, while making the appropriate substitutions when the values change. The stack trace includes the "source code", so that the state changes are paired with the appropriate reasoning steps on why the state needs to be changed in that fashion.

**Strengths:**

## potentially significant

This paper outlines several good prompting strategies, which are useful if you want to have the LLM to reason programmatically, with a rigid syntactic structure in its execution trace. The paper explains each strategy, irsa, skip attention, fragmented prompting with examples, and show the proposed prompts can achieve better results paired with a weaker model (gpt3.5) than a naive prompt with a more advanced model (gpt4)

The analogy of GPT as a turing machine is good too.

**Weaknesses:**

## poor quality and clarity

It is unclear if the proposed method can be reliably replicated to other domains, given that it is only evaluated on a handful of problems. I believe this work can be made substantially better if an automated method could be derived turning an existing complex program into a prompt, and evaluated on a larger set of problems, rather than the simplistic 100 python arithmetic problems.

## less than ideal novelty
It is also unclear how the proposed method is significantly different from Nye (2021)'s work on scratchpad, as both leverages trace information extensively. It would be good to have a related work section to spell out the exact differences.

The paper note that the proposed technique may be beneficial in the education domain. However, wouldn't having a LLM to simply mark up an existing execution trade of actually running the program be a better (and more correct) alternative?

**Questions:**

A program's execution can often have extraordinarily long traces, is the proposed method capable of handling this explosion of trace size, especially language models have a limited context window size? How would the proposed technique handle more complex program executions, which invariably have a long trace that would be infeasible to "print out" as sequences of tokens for an auto-regressive model?

---

> ### Author Response · Authors · 2023-11-23
>
> The LLM itself can be leveraged to create IRSA style prompts, as we did to create the LCS execution path using the interpreter/compiler Prompt A.2. Additionally, turning an existing complex program into a prompt could be achieved by performing an execution trace that adds in an IRSA style description (why then what) of the steps the program takes, especially conditions for entering or not entering control structures. This added explanation would in turn also describe conditions for continuing or stopping execution.
>
> Nye’s scratchpad work differs in that they tune their model specifically to perform the tasks and even in the proposed prompting experiment, they include a simple execution trace without IRSA style explanations, meaning that even if the trace information is there, the reasoning is not. Our comparison, simply adding an IRSA style explanation between line and state in the trace, forces the model to unroll its reasoning in addition to the trace and improves the accuracy from ~55% to ~90%. We will go into more detail in the final version of this paper to more clearly delineate the differences.
>
> Regarding explosion in trace length, skip attention and fragmented prompting can be used to address this issue, although newer models also have been increasing context window size and total token amounts. Skip attention reduces the dependency on token limit by allowing the model to continue generating from a state it has previously reached without attending to the text used to generate that state, and reduces the sensitivity to a problem with overlapping patterns illustrated in Fig. A.2. Fragmented prompting allows for exponentially reducing the prompt length (Prompt 3 discussed in Sec. 3), allowing the prompt to show state transitions on independent problems without following a full trace for a potentially very complex problem. Combining these two methods could allow IRSA prompting to be generalized to more complex program executions.

---

> > ### Comment · Reviewer_yv3R · 2023-11-23
> > **thanks for the response**
> >
> > thanks for the response, I am keeping my score as the response does not contain enough information to change my initial assessment of the paper substantially

---

> > > ### Author Response · Authors · 2023-11-23
> > >
> > > Re nye comparison, note that table 3 shows significant improvement on the type of programs they themselves proposed for evaluation. The appendix contains ablation studies (with an additional experiment now in Fig A.1 in the revision) that helps explain why IRSA outperformed Nye et al in Table 3.

---

### Official Review · Reviewer_RTek · 2023-11-02

**Soundness:** 2 fair
**Presentation:** 2 fair
**Contribution:** 2 fair
**Rating:** 3
**Confidence:** 3

**Summary:**

The paper proposes a highly structured prompting technique, IRSA, to trigger iterative (multi-step) logic execution in LLMs. The main contributions of the paper are algorithmic and empirical with some exploration of the implications to models of computation. In particular, the paper demonstrates that highly-structured prompts can be used to trigger LLMs to correctly simulate a "trace" (internal state, looping) of classical iterative algorithms (e.g., sorting). Experiments evaluating GPT's ability to simulate the execution of such algorithms shows that it does better on these computational tasks using IRSA prompts compared with less structured prompts.

**Strengths:**

+ The paper explores an important question of broad interest to the community. The exact capabilities and limits of LLMs remain unclear. This work shows that highly structured prompts can be used to better control LLM output on tasks requiring precise state control (memory) and iterative execution (loops).

+ The proposed prompts (IRSA) are intuitively clear. They seem novel, to my knowledge.

+ The experiments demonstrate that the IRSA prompts do indeed help the LLM correctly simulate algorithms requiring loops over the distribution of inputs considered. There is a good amount of detail included in the main paper and appendices.

**Weaknesses:**

- The use of a trace in the prompt raises a few questions, which are not addressed. A classic sorting algorithm can correctly sort very long lists using a relatively short specification of the algorithm. Can IRSA do the same (sort long input lists with a short trace)? The experiments seem restricted to short inputs length 5 in sorting, for example). The scaling, generalization and robustness of the IRSA prompt to different inputs aren't well explored in the paper.

- The paper could better highlight its algorithmic and empirical contributions relative to a rapidly growing body of literature on how to improve a LLM's instruction-following abilities. At the moment, I'm not sure if the experiments conclusively demonstrate that IRSA improves the instruction-following (via algorithm execution) abilities of a LLM.

- The terminology used can sometimes be a bit loose. For example, "this strategy hardens the attention", "skipping unnecessary attention saves computation", etc. More formal descriptions of these important ideas would increase the technical rigor of the paper. Alternatively, the paper could simplify the description to emphasize the empirical aspects (i.e., prompt engineering), which are also valuable.

**Questions:**

- How well do the proposed prompts do on significantly larger inputs? How does performance (esp correctness) vary with input size?

- How does the performance of the approach change if the LLM **hasn't** seen traces of the algorithms in its training data or not? For example, it seems plausible that traces of sorting and classical algorithmis might appear in training data. Does IRSA work if the trace is "new" to the LLM? Might synthetic tasks be needed here to eliminate this threat to validity?

- I was able to verify that GPT-4 correctly outputs a trace for the example shown in Prompt 1. However, the same prompt on a slightly larger input produces a correct answer but starts to include comments like one might see in code. Is this expected?

```
<original Prompt 1>
Problem: 0, 3, 3, 1, 2, 10, 98, 2000, 1232, 454422, 001, -222, 4533, 24, 99
EXECUTION
```

starts to produce comments like these

```
    // The iterations continue until no swaps are made in an entire iteration.

    Iteration:
       set swap_flag=false.
       // Comparisons occur here for all pairs.
       // ...

       // After several iterations, the list is eventually sorted.
       // No swaps occurred so, swap_flag=false
```

---

> ### Author Response · Authors · 2023-11-22
>
> Prompt 1 you tried out is less structured, performing worse than prompt A.4 (Table 1). On your problem, with Prompt A.4 GPT-4 follows its style at 0 temperature (did you try at 0?) without inserting comments. However, GPT-4 does shorten the answer after a few dozen steps and jumps to the conclusion, likely due to constraints that OpenAI has imposed (through tuning or brute force) to limit the number of tokens generated by the API/Playground at high comp cost (GPT-3.5-turbo-16k happily goes forever on this prompt).
>
> For longer traces, such as the ones in LCS, we introduced skip attention, where the model is forced to ignore all generated tokens except the last generated state. This approach is likely to work better for the purpose of algorithm execution than the techniques for extending the attention span generally in the models themselves. With skip attention, LCS prompts generated much longer token streams.
>
> Re. scaling, generalization, and robustness, a lot of it can be found in (now updated) Sec A.3.2, with an additional experiment in Figure A.1. IRSA style is meant to override model’s own “(over)thinking” and uncertainty, and force it to follow a recipe closely. Otherwise, any generated token that strays from the recipe can lead to at best unnecessary comments as in your experiment, and at worst “hallucination” leading to the wrong answer. In the main we illustrated the importance of IRSA with experiments with different bubble sort prompts, as well as on comparison with scratchpad traces. The appendix further studies the details, including in the new experiment in Fig. A.1. as well as the previous experiment in the section.
> Fig. A.1 studies evaluation of expressions of the form (a+-b)%10+c, which are hard for 2 reasons, impeding scratchpad prompts on random programs. 1st, the model often incorrectly computes modulo of a negative number, thinking that -47 % 10 equals -7. 2nd, multiple computations must be combined. But prompting with worked-out examples may not work perfectly without following the IRSA recipe. We look at the difference between prompting in “guess then check” style
>
> (24-41)%10+2=5
>
> 24-41=-17
>
> -17%10=?
>
> -17 is negative so the answer is 10-7=3
>
> -17%10=3
>
> (24-41)%10+2=5
>
> (plus another example with a positive result in a+-b)
> vs the “reason than answer” style that IRSA would follow, differing only in replacing the first 5 with a question mark.
> The only difference is one character per example (5 vs ? in the first line), but as the fig shows, the performance is very different. One might assume that with the first prompt, when working on a new expression, the model should guess correctly following the examples, and then even if wrong, it would have a chance of correcting itself at the end, as its generated explanation may change its mind. But the model often does not guess correctly as the reasoning requires iterative attention that it can't do without unrolling the reasoning step by step. Even after generating a correct follow up step-by-step reasoning and computing correctly (a+b)%10, it has trouble deciding if it should add c or follow its initial guess. The 2nd prompt doesn't create this uncertainty as the model follows the recipe and generates a placeholder ? instead of an initial guess, which later doesn't conflict with the generated reasoning. The answer entropy quickly rises with temperature (even for GPT-4) for the 1st prompt, but not for the 2nd. Temperature is a way to artificially increase uncertainty of generation, but due to the size of the training set and possible contradictions in it, uncertainty (epistemic) is inherent to models even at 0 temp. IRSA is meant to instruct in a way that reduces this uncertainty and increases accuracy.
>
> Re. the focus on prompting strategy vs algorithmic contributions, we agree that both are important and we will discuss them as such in the final version. Briefly on the prompt engineering front, there are issues like discussed above and in Secs 2 and A.3.2. The algorithmic contributions are skip-attention which reduces the dependency on token limit and reduces the sensitivity to a problem illustrated in Fig. A.2; and fragmented prompting, which allows for exponentially reduced the prompt length (Prompt 3 discussed in Sec. 3)
>
> Re. the execution traces in the training data:
> For classical algorithms, the traces were in the training data (try 0-shot Prompt A.10 with higher temp a few times), yet without IRSA, 0-shot code execution doesn't work anywhere near as well (GPT-4 gets only 69% compared to 93% for GPT-3 w/ IRSA, see Sec A.3.3).
> Table 3 shows results on new random programs we synthesized, for which traces couldn't have been in the data. Thus, having examples of traces in training likely helps GPT interpret programs, but this is not limited to programs it saw in training. Our experiments show that generalization to new programs gets much better with IRSA, and extension to longer programs benefits greatly from skip attention and fragmented prompting

---

### Meta-Review · Area_Chair_sWB9 · 2023-12-04

**Metareview:**

The authors present an interesting prompting strategy that causes LLMs to simulate algorithms through in-context learning. Conceptually it is quite similar to scratch pads / chain of thought. The primary weakness of the paper is that it is not very novel compared to those earlier works, and the experimental evaluation is limited to the point of being borderline anecdotal. It would be much stronger paper if it showed that this method could somehow be adapted to a nontrivial subset of tasks on bigbench, for example.

**Justification For Why Not Higher Score:**

The weakness of the experimental evaluation and relative lack of novelty--More broadly, IMO, a primary contribution of introducing a new prompting trick demands a high bar for publication.

**Justification For Why Not Lower Score:**

n/a

---

### Decision · Program_Chairs · 2024-01-16

Reject